# Temporal lineage replacements and dominance of imported variants of concern during the COVID-19 pandemic in Kenya

Gathii Kimita [1], Josphat Nyataya[1], Esther Omuseni[1], Faith Sigei [1], Alan Lemtudo [1], Eric Muthanje[1], Brian Andika[1], Rehema Liyai[1], Rachel Githii[1], Clement Masakwe [1], Stephen Ochola[1], George Awinda[1], Carol Kifude[1], Beth Mutai [1], Robert M. Gatata[2] & John Waitumbi [1✉]

## Abstract

**Background**  Kenya's COVID-19 epidemic was seeded early in March 2020 and did not peak until early August 2020 (wave 1), late-November 2020 (wave 2), mid-April 2021 (wave 3), late August 2021 (wave 4), and mid-January 2022 (wave 5).

**Methods**  Here, we present SARS-CoV-2 lineages associated with the five waves through analysis of 1034 genomes, which included 237 non-variants of concern and 797 variants of concern (VOC) that had increased transmissibility, disease severity or vaccine resistance.

**Results**  In total 40 lineages were identified. The early European lineages (B.1 and B.1.1) were the first to be seeded. The B.1 lineage continued to expand and remained dominant, accounting for 60% (72/120) and 57% (45/79) in waves 1 and 2 respectively. Waves three, four and five respectively were dominated by VOCs that were distributed as follows: Alpha 58.5% (166/285), Delta 92.4% (327/354), Omicron 95.4% (188/197) and Beta at 4.2% (12/284) during wave 3 and 0.3% (1/354) during wave 4. Phylogenetic analysis suggests multiple introductions of variants from outside Kenya, more so during the first, third, fourth and fifth waves, as well as subsequent lineage diversification.

**Conclusions**  The data highlights the importance of genome surveillance in determining circulating variants to aid interpretation of phenotypes such as transmissibility, virulence and/or resistance to therapeutics/vaccines.

### Plain language summary

The SARS-CoV-2 virus that causes COVID-19 has been changing over time. We investigated the changes in the virus present in Kenya from March 2020 to January 2022. During this period there were five successive waves of infection, during which time more people were infected with the virus. The virus arrived in Kenya from other countries many different times. Identifying the changes in the virus helps us to understand how the virus changes over time and the effect this has on its ability to infect people and make them ill.

---

[1] Kenya Medical Research Institute/US Army Medical Research Directorate–Africa/Kenya, Basic Science Laboratory. Off Kisumu - Kakamega Highway, Kisumu, Kenya. [2] Ministry of Defence, Nairobi, Kenya. ✉email: john.waitumbi@usamru-k.org

Severe acute respiratory syndrome coronavirus 2 (SARS-CoV-2) has impacted public health, social, political and economic spheres of life since its emergence in Wuhan, China and subsequent spread to the rest of the world. It reached every part of the globe in less than 9 months, and at the time of writing this report, it had infected over 535 million and killed over 6,309 million people globally. The virus is the etiological agent of coronavirus disease 2019 (COVID-19), a mysterious severe respiratory illness that first appeared in Wuhan, Hubei Province, China, in December 2019[1]. The origin of SARS-CoV-2 is controversial[2] but from genetic studies, the closest relatives are bat coronaviruses[3–6]. It remains to be proven whether the bat facilitated both the evolution of SARS-CoV-2 and its transmission to humans[7]. A recent study examining the evidence for a spillover event from Wuhan versus a Chinese Lab, found overwhelming evidence for a market in Wuhan as the most probable source of SARS-CoV-2, and not a Chinese government laboratory. According to that study, two independent zoonotic spillover events occurred two weeks apart, the first involving lineage B viruses while the second involved lineage A[8]. To date, COVID-19 has overshadowed all other human health calamities, ravaged global economies and disrupted human social interactions[9]. With the spread to different countries, the original A and B variants have diversified leading to the emergence of multiple variants, some with greater virulence than others[8].

In Kenya, the first confirmed case of COVID-19 was on 12th March 2020 from a Kenyan citizen returning home from the USA via London, UK[10]. Within two weeks, 31 cases traceable to the index case and other international travelers were identified. To try and curtail the spread, the government instituted a series of countermeasures that included border closures, mandatory quarantine on returning travelers, night curfews, ban on gatherings, and mandatory mask use while in public spaces[11]. While these measures slowed the spread of the disease, the virus still managed to infiltrate into the community, and new infections associated with local transmission events continued to drive the spread. To date, Kenya has reported over 327,000 cases and 5000 deaths[12], but the total number of cases are likely to be a gross underestimate considering the inadequacy of testing[13,14].

The earliest published description of SARS-CoV-2 in Kenya covered the period between February and March 2021 and traced the introduction and spread of European lineages in the coastal region of Kenya[15,16]. In the whole country, wave one peaked in early-August 2020, and by mid-September 2020, SARS-CoV-2 numbers had declined to very low levels, marking the "end" of the first wave. Buoyed by the reduction in COVID-19 numbers, most of the COVID-19 restrictions were lifted to ease pressure on a slumping economy[17]. The next one and half months were characterized by low infections, but this short-lived lull was interrupted by a spike in the number of cases that rose steadily, peaking at 1,554 by mid-November 2020. This triggered another round of lockdowns. The reopening of schools was postponed, and public gatherings, especially political rallies, stood banned. These measures, and probably because of other reasons progressively reduced infections, and by January 2021, the second wave burned out. Using population mobility models, it was theorized that the second wave was triggered by a population of Kenyans associated with higher socialeconomic status returning to pre-COVID-19 mobility patterns[18].

Between January 2021 and February 2022, Kenya experienced three COVID-19 waves, all of which were associated with SARS-CoV-2 variants of concern (VOCs) and variants of interest (VOIs) that emerged independently from different parts of the globe. The VOCs were associated with increased transmissibility, virulence, clinical presentation and decreased effectiveness to diagnostics and therapeutics, including vaccines[19]. Kenya has so far detected 4/5 currently classified VOCs, i.e. the Alpha variant (B.1.1.7/ 20I/V1), Beta (B.1.351, 20H/V2), Delta (B.1.617.2, 21 A/21I/21 J)[18,20] and Omicron (B.1.1.529, 21 K/21 L/21 M)[21]. SARS-CoV-2 genomes deposited in GISAID and Genbank from Kenya indicate the presence of two VOI, i.e. Eta (B.1.525, 20 A/S484K) and Kappa (B.1.617.1, 21B). The initial fear that some of the emerging mutants could negatively impact vaccine efficacy and constitute postvaccination "antigenic escape" has been witnessed with the Delta[22] and Omicron[23].

In this report, we use genomic surveillance to dissect the five COVID-19 waves that have occurred in Kenya since the beginning of the outbreak and provide data that show temporal lineage dominance, diversification and emergence of the more transmissible VOCs.

## Methods

**Ethics statement**. This study was performed as part of public health surveillance approved by the Kenya government through the Ministry of Health (MOH) as part of response to the COVID-19 pandemic. The Scientific and Ethical Research Unit (SERU) of Kenya Medical Research Institute (KEMRI) approved a country-wide protocol to allow SARS-CoV-2 whole genome sequencing (SERU 4035) for all samples that were being collected as part of COVID-19 pandemic response, in order to allow tracking of virus evolution. Since the samples were obtained as part of public health surveillance, the IRB also waived the need to obtain prior informed consent.

**Study sample acquisition**. Multiple laboratories, including the Basic Science Laboratory (BSL) in Kisumu were designated by the MOH as COVID-19 testing centers. BSL started supporting COVID-19 mass testing and whole genome sequencing in March 2020, and by February 2022, the laboratory had screened 63,542 respiratory samples by RT-qPCR. Samples came from different parts of the country, and where indicated, a few came from other countries (Table 1). Nucleic acid isolation was performed using MagMAX Viral/Pathogen nucleic acid isolation Kits with the KingFisher Flex particle purification system (Thermo Fisher

**Table 1 Demographic data for subjects who contributed genome sequences used in this study.**

|  |  | N | % |
|---|---|---|---|
| **Age distribution (years)** | <20 | 83 | 8.0 |
|  | 21-30 | 268 | 25.9 |
|  | 31-50 | 404 | 39.1 |
|  | 51-60 | 83 | 8.0 |
|  | >61 | 89 | 8.6 |
|  | missing data | 107 | 10.3 |
| **Median age** | 35 |  |  |
| **Gender** | Male | 574 | 55.5 |
|  | Female | 357 | 34.5 |
|  | ? | 103 | 10.0 |
| **Nationality** | Kenya | 1017 | 98.4 |
|  | Uganda | 9 | 0.9 |
|  | Congo | 2 | 0.2 |
|  | India | 1 | 0.1 |
|  | Zimbabwe | 1 | 0.1 |
|  | Japan | 1 | 0.1 |
|  | USA | 2 | 0.2 |
|  | Rwanda | 1 | 0.1 |
| **Infection waves** | Wave 1 | 120 | 11.6 |
|  | Wave 2 | 79 | 7.6 |
|  | Wave 3 | 284 | 27.5 |
|  | Wave 4 | 354 | 34.2 |
|  | Wave 5 | 197 | 19.1 |

Scientific, CA, USA). Of the 63,542 samples tested, 8.5% ($n = 5375$) were positive for SARS-CoV-2 at varying cycle thresholds (Ct). Of these, 1089 with Cts <33 were selected for whole genome sequencing.

**Whole genome sequencing and genome assembly**. Complementary DNA (cDNA) was synthesized from RNA using random primers with either Superscript IV one-step reverse transcriptase kit (Thermo Fisher Scientific, CA, USA) or the LunaScript RT SuperMix Kit (New England Biolabs, MA, USA). The cDNA was then used for tiled multiplex PCR using the Q5 High-Fidelity 2X Master Mix (New England Biolabs) and ARTIC v3 primers (Supplementary Data 3 on Figshare) as described in the associated protocol[24]. The amplicons were cleaned with AMPureXP beads (Beckman Coulter, USA) and then used to create sequence libraries using the NexteraXT (Illumina, CA, USA) and Collibri ES (ThermoFisher Scientific, CA, USA) library preparation kits, as per the manufacturers' instructions. The libraries were assessed on D1000 HS screen tape on a Tape Station 4200 (Agilent, CA, USA) for size distribution and concentration. A 12 pM library spiked with 10% Phix genome were sequenced on the MiSeq benchtop sequencer (Illumina, CA, USA) using 600 cycles V3 paired-end chemistry.

Read demultiplexing was conducted onboard the MiSeq using the MiSeq reporter v2.6. The reads were quality filtered to remove Illumina adapters and low-quality sequences using Trimmomatic v0.36[25]. Trimmed reads were assembled against SARS-CoV-2 Wuhan 1 as a reference (GenBank accession number: NC_045512) using bwa 0.7.5[26]. Samtools v0.1.19[27] was used to create pileups from the alignment, while ivar v1.3.1[28] was used to remove primers and build the consensus sequence. The consensus sequences were further curated using Nextclade Web v1.35.1[29] to screen out samples with too much missing information (Ns > 30% of the genome), mixed sites and private mutations.

**Lineage and clade assignment**. Lineage assignment was performed on each consensus sequence using PANGOLIN v3.1.17 and PANGOLEARN v2021-12-06 (Phylogenetic Assignment of named Global Outbreak LINeages)[30], which offers a hierarchical dynamic nomenclature describing a lineage as a cluster of sequences observed in a geographically distinct region with evidence of transmission in that region. Clades were assigned to each consensus sequence using Nextclade Web v 1.13.1. The Nextstrain clade system[31] uses a year-letter nomenclature on a clade exceeding 20% global representation and >2 positional differences from its parent clade while considering clade persistence with time as well as the extent of its geographical spread. Of the 1089 COVID-19 nasal specimens that passed the threshold for whole genome sequencing (Cts <33), 45 were dropped because they did not pass the threshold required for assigning Pango lineages. Ten additional samples were dropped because they lacked date of collection. The remaining 1034 genomes were used to monitor the evolution of SARS-CoV-2 lineages across the five COVID-19 waves.

**Global data acquisition**. To compare the genome sequences of the current study to global sequences, SARS-CoV-2 genomes were sampled from the Global Initiative on Sharing All Influenza Data (GISAID)[32]. Due to the huge number of genomes present in the GISAID, we opted to utilize the globally sampled genomes maintained by Nextstrain. These genomes are sub-sampled using a criterion that considers their spatial–temporal characteristics, as well as their genotypes, to yield a balanced and inclusive sub-sample of 2891 genomes. We also downloaded all SARS-CoV-2 genomes from Kenya deposited in GISAID that were outside this

study, as well as very early reported genomes of each VOC. All the datasets were downloaded/sampled on 17th February 2022. For the Kenyan datasets and early reported genomes for VOCs, only sequences flagged as "complete (>29,000 bp)", "high coverage only" (entries with <1% Ns and <0.05% unique amino acid mutations not seen in other sequence databases and with no unconfirmed insertion/deletions), and "low coverage excl" (excluded entries with >5%Ns) were downloaded from GISAID.

**Phylogenetic analysis**. Of the 1034 genomes that were used for lineage assignments, only 969 with genome lengths >27000 bp could be used for phylogenetic analysis.

Five phylogenies were constructed; one to determine the phylogenetic placement of the study genomes against a background of globally sampled genomes (context genomes). This tree consisted of 316 context genomes sampled from around the globe and 969 genomes from this study. The Alpha variants tree was constructed with genomes from Kenya ($n = 381$), against a global subsample ($n = 164$) that included some of the earliest reported Alpha variants ($n = 43$), mostly from England. The Beta variants tree was constructed with genomes from Kenya ($n = 27$), against a global subsample ($n = 55$) that included early Beta variants from Southern Africa ($n = 30$). The Delta variants tree was constructed with genomes from Kenya ($n = 634$) and a global subsample ($n = 232$) that included the earliest reported Delta variants from India ($n = 22$). The Omicron variants tree was reconstructed from the Kenyan genomes ($n = 827$) and a global subsample ($n = 214$) that included the earliest reported Omicron variants, mostly from South Africa ($n = 8$).

All trees were reconstructed with augur v14 as implemented in the Nextstrain pipeline version 3.0.6[31]. Within Nextstrain, a random subsampling method was used to cap the maximum number of context sequences from the rest of the world – to provide phylogenetic context, based on genomic proximity. Only genomes >2700 nt long were aligned with nextalign v1.11[29]. Phylogenies were reconstructed using IQTree[33] employing a General Time Reversible (GTR) model. Estimation of time-scaled phylogenies was done using Tree Time v0.8.6[34], assuming a nucleotide substitution rate of $8 \times 10^{-4}$ per site per year, and a coalescent model. The resulting trees were visualized using auspice v2.29.1 and figtree v1.4.4[35].

**Reporting summary**. Further information on research design is available in the Nature Research Reporting Summary linked to this article.

## Results

Demographic data of subjects who contributed the 1034 genomes is shown in Table 1, and Supplementary Data 1 on Figshare. Of those with age data, most samples came from age groups 31–50 ($n = 404$) and 21–30 years ($n = 268$). 107 specimens did not have age data. More sequences were from males (55.5%, $n = 574$) than females (34.5%, $n = 357$), notwithstanding that 103 individuals (10.0%) did not indicate gender. For ease of analysis, we agglomerated Kenyan counties into regions. The Nyanza region contributed most of the genomes ($n = 337$), Nairobi metropolitan area ($n = 187$), Rift Valley ($n = 186$), Western region ($n = 137$), Central region ($n = 131$). The Coastal region contributed the least ($n = 56$). North Eastern and Eastern regions were unrepresented.

**Expansion and displacement of SARS-CoV-2 lineages and eventual dominance of the VOC**. Of the 1089 nasal specimens that passed the threshold for whole genome sequencing (Cts <33), 1034 collected between May 2020 and January 2022 had useable genomes and were used to monitor the evolution of SARS-CoV-2

lineages across the five COVID-19 waves. The waves were defined by increase and decrease of positive cases over time and in general corresponded to the waves observed in the whole country (Supplementary Fig. 1). Of the 1034 genomes, 237 were non-variants of concern and 797 were VOCs.

By Nextclade nomenclature, 13 clades were identified from the 1034 study genomes and are shown in the Supplementary Fig. 2: 21A-Delta contributed 298 genomes (28.8%), 192 from 20I-Alpha V1 (18.6%), 188 from 21K-Omicron (18.2%), 98 from 20C (9.5%), 94 from 21J-Delta (9.1%), 60 from 20A (5.8%), 34 from 20B (3.3%), 24 from 19B (2.3%), 19 from 21D-Eta (1.8%), 13 from 20H-Beta, V2 (1.3%), 12 from 21I-Delta (1.2%) and two from 20D (0.2%). By Pango lineage, these genomes delineated into forty distinct lineages, nine of which were linked to wave one (collected from May 2020 to mid-September 2020), 11 to wave two (late-September 2020 to mid-January 2021), 12 to wave three (January to early-June 2021), 17 to wave four (June to mid-November 2021), and 5 to wave five (Mid-November 2021 and January 2022) (Supplementary Table 1). Figure 1 shows the relationships between Pango lineages across the five COVID-19 waves. Despite the presence of multiple variants in each wave, each had a set of dominant lineages (Fig. 1A), that were sometimes shared across the waves, but in general, shared lineages were fewer (Fig. 1B). The nine lineages identified in wave one came from 120 genomes. The lineages in this wave were mainly of European descent, with B.1 accounting for 60.0% of the lineages, followed by B.1.1 (22.5%) (Fig. 1A). Ugandan lineage A.25, accounted for 6.7%. Other minor lineages were from the USA (B.1.243, 2.5%), Kenya/Uganda (B.1.393, 2.5%), Kenyan (B.1.549, 1.67%), England (B.1.1.1, 1.67%) and USA (B.1.1.356, 1.67%), while the original haplotype of the pandemic (lineage A) had a frequency of 0.81%. The eleven lineages identified in wave two came from 79 genomes. All but B.1 and B.1.549 were new (i.e. not detected in the first wave). B.1 remained as the dominant lineage (57.0 %), while the Kenyan lineage, B.1.549, which was a minor lineage (1.67%) during wave 1, became the second most dominant (13.9%), followed by another Kenyan lineage, B.1.530 (10.1 %). Other minor lineages (<5%) were from East Africa (A.23.1), Kenya (B.1.596.1), Uganda (A.23), South Africa (B.1.1.254), Denmark (B.1.428), USA (B.1.340 and B.1.596) and Kenya (N.8).

During wave three, 12 lineages were identified in 284/1034 (27.5%) genomes. It is during wave three that VOCs and VOI began to emerge in Kenya: the Alpha (B.1.1.7) originally identified in the UK, Beta (B.1.351) originally identified in South Africa and Delta originally identified in India (B.1.617.2), as well as the Eta VOI (B.1.525), whose earliest sequences are linked to West Africa lineages. Collectively, the VOC/VOI expanded rapidly, replacing almost all wave two lineages to account for 93.3% of all lineages. The Alpha variant was the most dominant during wave three at a frequency of 58.45%, while the Delta variant and its sub lineages collectively accounted for 24.0% (the AY.16 being the dominant Delta sub-lineage accounting for 65/68 genomes). The Eta VOI had a frequency of 6.7 %, while the beta VOC had a frequency of 4.2%. Remnant variants from previous waves occurred at a diminishing rates: A.23.1 (3.8 %), B.1 (1.4 %), B.1.1, B.1.530 and B.1.12 at <1%.

During wave four, 17 lineages were identified in 354 genomes, 92.4% of which were Delta lineage: AY.16 at 61.3%, AY.46 at 17.2%, AY.116 (1.9%), AY.122 (1.7%), AY.46.4 (1.7%), AY61 (1.4%), AY.65 (1.4%), and AY.71, AY.120.2, AY.109, AY.126, AY.16.1, AY.41 at <1%. Infection rates with the Alpha variant that had dominated wave three diminished to 7.34%. The beta variant was also detected at a frequency of <1%.

During wave five, 5 lineages were identified in 197 genomes. This wave was totally dominated by Omicron (95.4%), with BA.1

and BA.1.1 lineages occurring at a frequency of 13.2% and 82.2%, respectively. Remnant Delta sub-lineages (AY.46 at 3.5%, and AY.16 and AY.116 occurring at 0.5%) were also identified.

**Time-scaled phylogenetic tree of Kenyan samples**. In order to contextualize the study genomes to the global phylogenetic temporal scale, a time-scaled phylogenetic tree that included 316 genomes sampled from around the globe and 969 from this study was constructed (Fig. 2). Kenyan genomes branched into multiple lineages, suggesting multiple seeding events. They also formed monophyletic clusters with notable intercluster divergence, indicating local transmission and diversification.

**Emergence and dominance of VOCs**. While the whole of 2020 was dominated by the European and local lineages of SARS-CoV-2, the subsequent year was dominated by VOCs.

*The Alpha VOC*. A time-scaled phylogeny involving 546 B.1.1.7 genomes, 381 from Kenya (182 from our study) and 165 sub-sampled from the globe is shown in Fig. 3. The tree is rooted against the Wuhan/WHO1/2019 and Wuhan/Hu-1/2019 genomes (Genbank accession no. LR757998 and MN908947 respectively). The genomes from the study are shown in dark blue circular tips, while other Kenyan genomes are shown in light blue tips; date range: 18 January 2021 to 26 November 2021. Genomes from other parts of the world (shown in yellow tips) include 44 genomes sampled from the earliest reported alpha variants; date range: 24 October 2020 to 01 December 2021, most from England. The Kenyan genomes branched from different parts of the tree, indicating multiple independent alpha variant seeding events. The first reported outbreak involving Alpha was from samples brought into the laboratory in February 2021 from an outbreak cluster that occurred in Nanyuki, a small town in Laikipia County at the foothills of Mount Kenya. The suspected sources were British soldiers returning from the United Kingdom. All the associated samples were of the B.1.1.7 lineage. The samples branched at three different locations on the tree (Fig. 3, black arrows), indicating the introduction of three different variants; of these, only one source was responsible for the large outbreak cluster (n = 32). From our analysis, the earliest introduction of the Alpha variant in Kenya outside the Nanyuki outbreak is indicated by two genomes (Kenya/SS2930/2021 and Kenya/SS2927/2021) shown in encircled red dots, collected in Nairobi on 18 January 2021. Both samples were from unlinked sources as they branched from different parts of the trees.

*The Beta VoC*. A time-scaled tree consisting of 82 B.1.351 genomes, 27 from Kenya (12 from this study) and 55 sub-sampled from around the world is shown in Fig. 4. The tree is rooted against the Wuhan/WHO1/2019 and Wuhan/Hu-1/2019 reference genome (Genbank accession no. LR757998 and MN908947, respectively). Genomes from the current study are shown in dark blue circular tips, while other Kenyan genomes are shown in light blue circular tips; date range: 16th February 2021 to 30th July 2021. Genomes from other parts of the world are shown in yellow tips (n = 55), and included early Beta variants from South Africa collected between 4th and 30th September 2020 (n = 29). South African genomes occurred in the basal parts of the tree, while majority of the Kenyan beta variant genomes were in the more derived parts of the tree. The earliest beta genome from the study samples was a sample from Lamu Island, coastal Kenya (AFI-LAM-130, gisaid epi_isl_2779301) collected on 16th February 2021.

*The Delta VoC*. A time-scaled phylogenetic tree involving 836 B.1.617.2 genomes rooted with the Wuhan/WHO1/2019 and Wuhan/Hu-1/2019 genome references (Genbank accession nos.

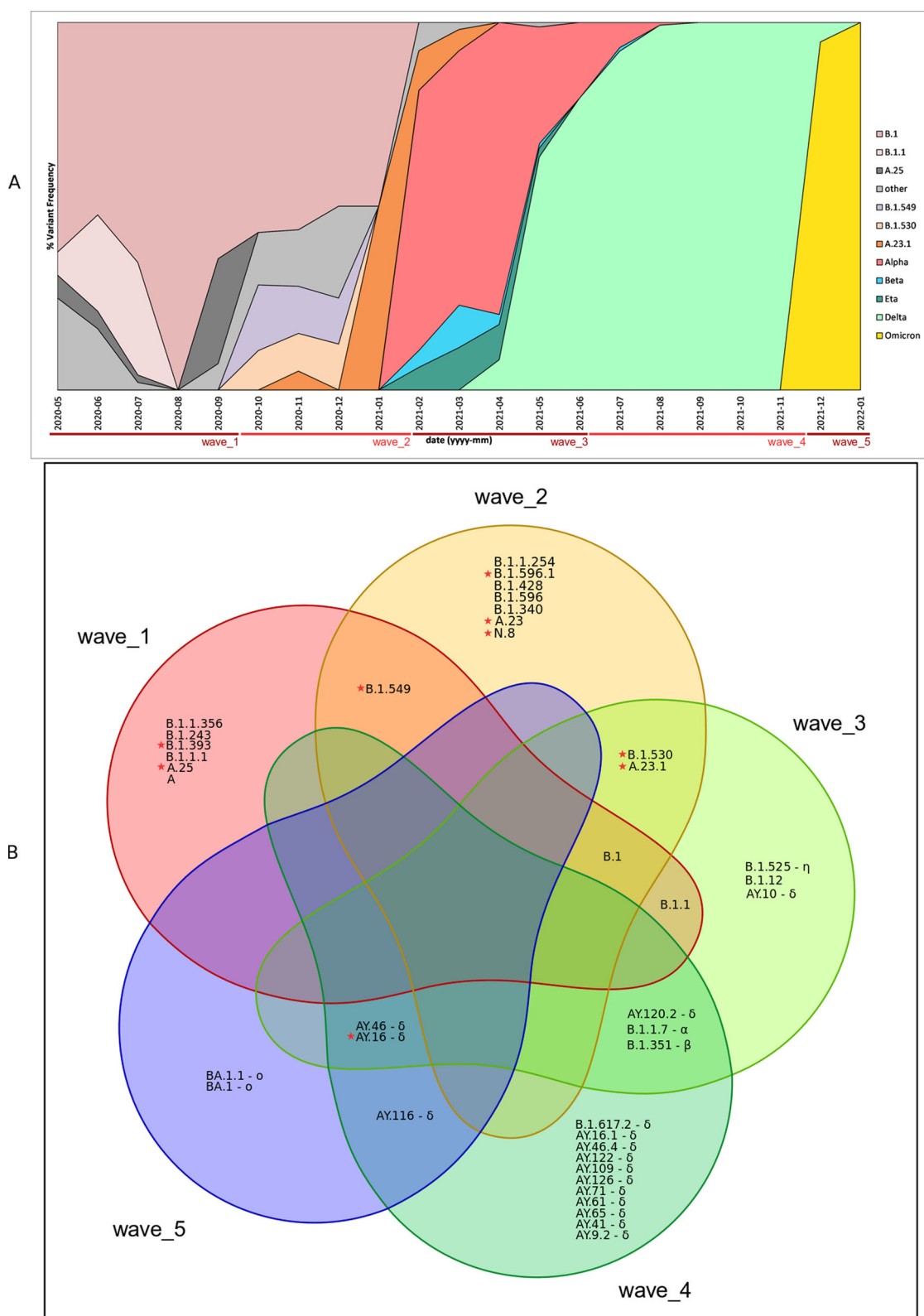

LR757998 and MN908947 respectively) is shown in Fig. 5. The tree consists of 605 Kenyan samples (396 from our study) and 231 subsampled from around the world. Genomes from this study are shown in dark blue circular tips, while other Kenyan genomes are shown in light blue circular tips; date range: 8th April 2021 to 17th December 2021. 231 genomes from other parts of the world are shown in yellow circular tips, and include the

earliest reported Delta variants (date range: 05 February 2021 to 31 March 2021), from India ($n = 29$). In the tree, Kenyan samples branched mainly with AY.16 ($n = 413$, grey branches) and AY.46 ($n = 135$, black branches). The earliest introductions were two samples Kenya/ILRI_COVM01231/2021 – lineage AY.16 (gisaid isl no. EPI_ISL_6096022) and Kenya/ILRI_COVM01217/2021 - lineage Ay.61 (gisaid isl no EPI_ISL_6095984) encircled in red

**Fig. 1 SARS-CoV-2 lineages circulating across the five COVID-19 waves in Kenya.** SARS-CoV-2 lineages circulating across the five COVID-19 waves in Kenya. Panel **A** shows monthly distribution of the major SARS-CoV-2 pango lineages and/or variant of concern/interest identified between May 2020 and January 2022. Red holizontal bars indicate the associated waves. Imported European lineages B.1 and B.1.1 dominated in waves one and two, although by wave two, there was an expansion of local lineages such as B.1.549, B.1.530 and A.23.1. Wave three was dominated by the Alpha VOC which progressively displaced the variants identified in the previous two waves. Wave four was the longest (lasting 5 months), and was dominated by the Delta VOC. Wave five, caused by the hyper transmissible Omicron variant, emerged stealthily, quickly spread and replaced Delta and the remnant Alpha variants, to become the dominant VOC. Panel **B** is a Venn diagram showing the relationships between the 40 Pangolin lineages across the five COVID-19 waves. Each wave had characteristic dominant lineages; some shared across the waves, but in general, the shared lineages were fewer. Red stars denote local lineages. Greek symbols denotes VOC or VOI (α = Alpha, β = Beta, δ = Delta, η = Eta, o = Omicron).

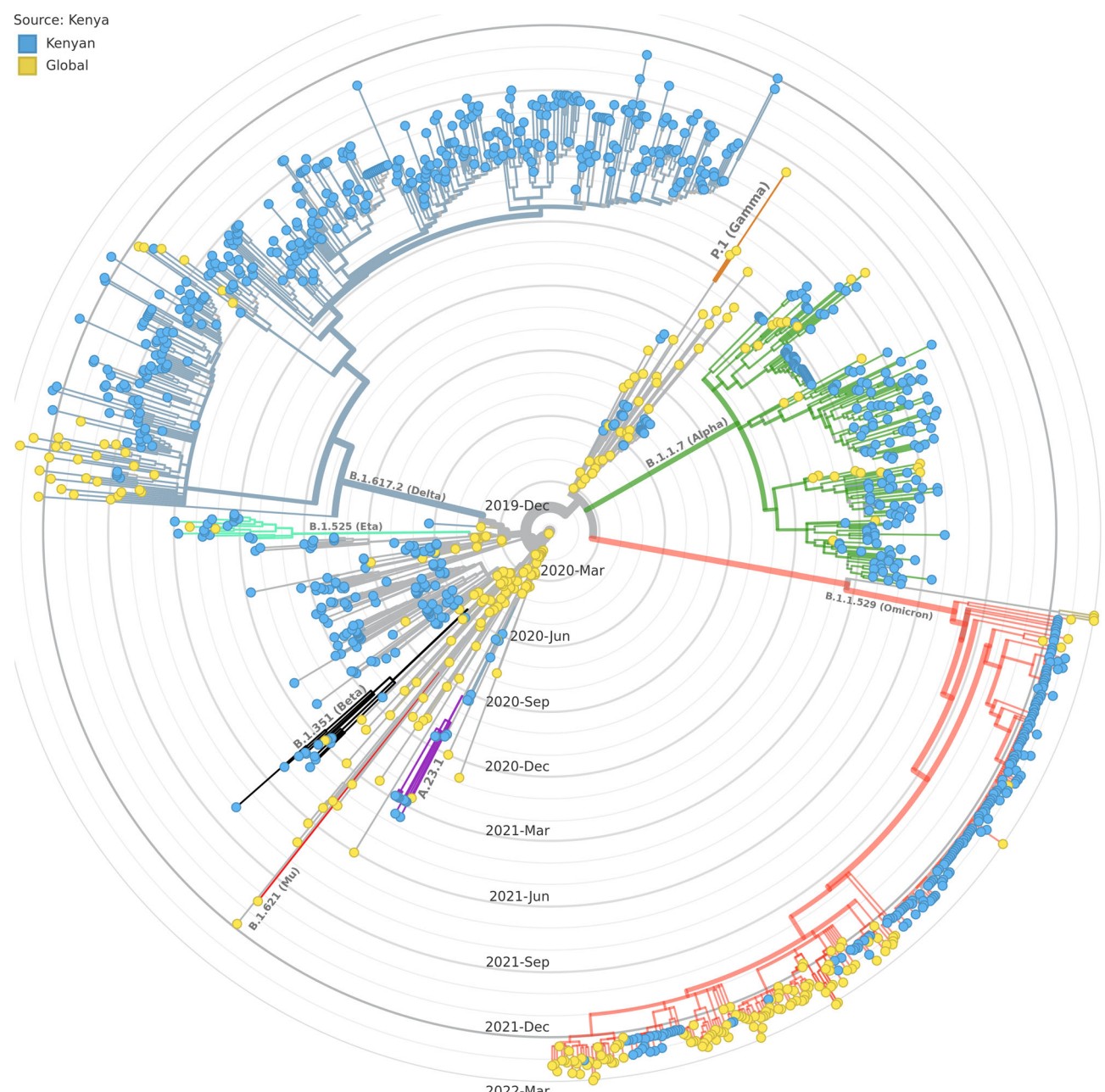

**Fig. 2 Time-scaled phylogenetic tree of Kenyan genomes against selected global genomes.** The tree was constructed with 316 genomes sampled from GISAID and 969 genomes from this study. The study samples were resolved into non-VOC/VOI lineages (grey branches), Delta (blue branches), Omicron (red branches), Alpha (green branches), Beta (black branches), Eta (teal branches) and the A.23.1 lineage (purple branches). No Kenyan samples branched with either the gamma variant (orange branches) or the Mu variant (crimson branches). Yellow tips represent global genomes, while blue tips represent the study samples. The different colors on branches represent the Pango lineages.

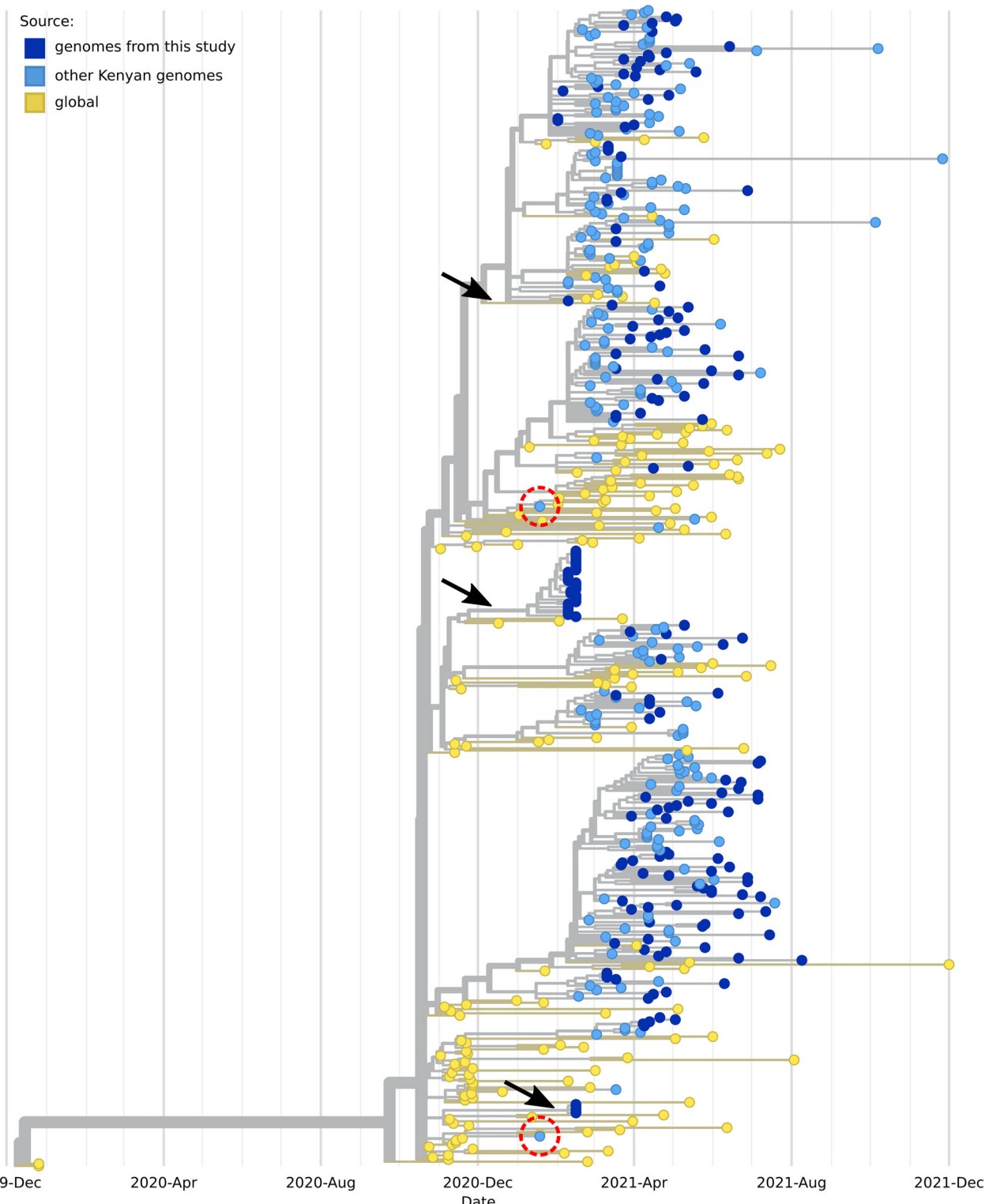

**Fig. 3 Phylogenetic tree of the B.1.1.7 lineage from our samples and those sub-sampled globally.** The tree was constructed with 546 genomes, 381 from Kenya (182 from our study) and 165 sub-sampled from the globe. Genomes from other parts of the world include 44 genomes sampled from the earliest reported Alpha variants. The tree is rooted against the Wuhan/WHO1/2019 and Wuhan/Hu-1/2019 reference genomes (Genbank accession no. LR757998 and MN908947, respectively). The genomes from the study are shown in dark blue circular tips while other Kenyan genomes are shown in light blue circular tips. Genomes from other parts of the globe are shown in yellow circular tips. Kenyan samples branched from different parts of the tree, indicating multiple independent seeding events. Black arrows show samples from one of the first reported outbreak of the B.1.1.7 lineage in Kenya. Encircled red dots indicate earlier two independent introductions (18 January 2021) of the alpha variant outside the Nanyuki outbreak.

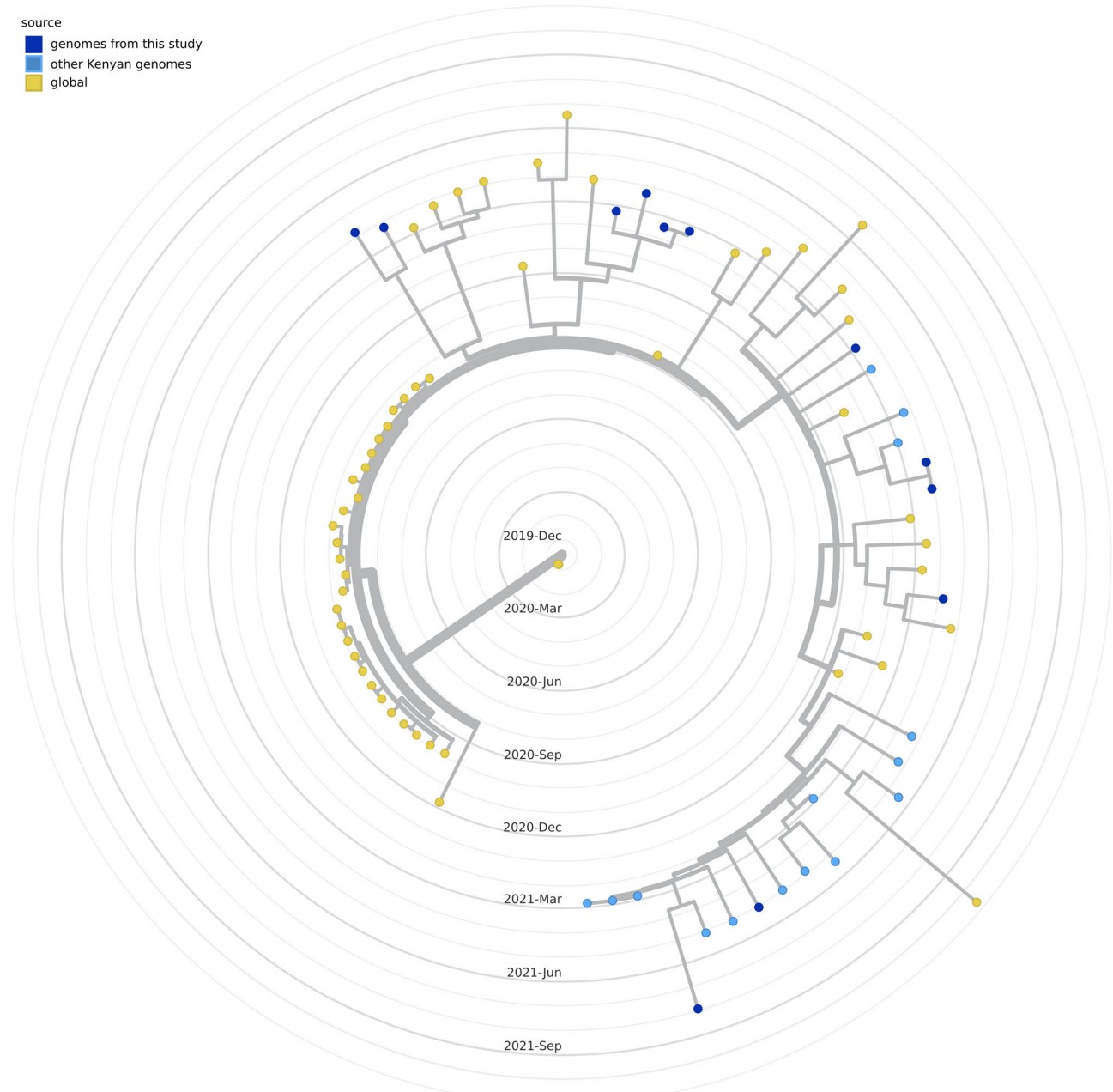

**Fig. 4 Phylogenetic tree of the B.1.351 lineage from Kenyan samples and those sub-sampled globally.** The tree was constructed from 82 genomes - 27 from Kenya (12 from this study), and 55 sub-sampled genomes from around the world. The global sub-sample included 29 genomes sampled from the earliest reported beta variants. The tree is rooted against the Wuhan/WHO1/2019 and Wuhan/Hu-1/2019 reference genome (Genbank accession no. LR757998 and MN908947). The genomes from this study are shown in dark blue circular tips, while other Kenyan genomes are shown in light blue circular tips. Genomes from other parts of the globe are shown in yellow circular tips. South African genomes occurred in the basal parts of the tree, while majority of the Kenyan genomes were in the more derived parts of the tree.

and black, respectively, both collected from Nairobi on 8th April 2021.

*The Omicron VoC*. A time-scaled phylogenetic tree involving 1039 B.1.1.529 genomes rooted against the Wuhan/WHO1/2019 and Wuhan/Hu-1/2019 SARS-CoV-2 reference genomes, genbank accession identifier LR757998 and MN908947 respectively is shown in Fig. 6. The tree comprised 827 Kenyan samples (188 from this study) and 214 sub-sampled from the globe. The genomes from the study are shown in dark blue circular tips while other Kenyan genomes are shown in light blue circular tips; date range: 27th November 2021 to 15th January 2022. Genomes from other parts of

the world (shown in yellow tips) include 8 genomes sampled from the earliest reported omicron variants; date range: 19th November 2021 to 30th November 2021, most from South Africa. Majority of the Kenyan samples branched with the (BA.1 lineage). Within this lineage, there is evidence of diversification as the genomes delimit into three sub-clusters (Fig. 6A–C), with majority of the Kenyan samples (*n* = 651, 62.5%) branching in the more derived sub-cluster. Of these, 620 were pango lineage BA.1.1. Only one sample branched with the BA.2 lineage Kenya/ILRI_COVM01771/2021 (GISAID identifier EPI_ISL_9093518). The first introductions of Omicron were in 27 November 2021 from four samples collected in Nairobi (Fig. 6, light blue tips with red borders) and all were of BA.1 lineages

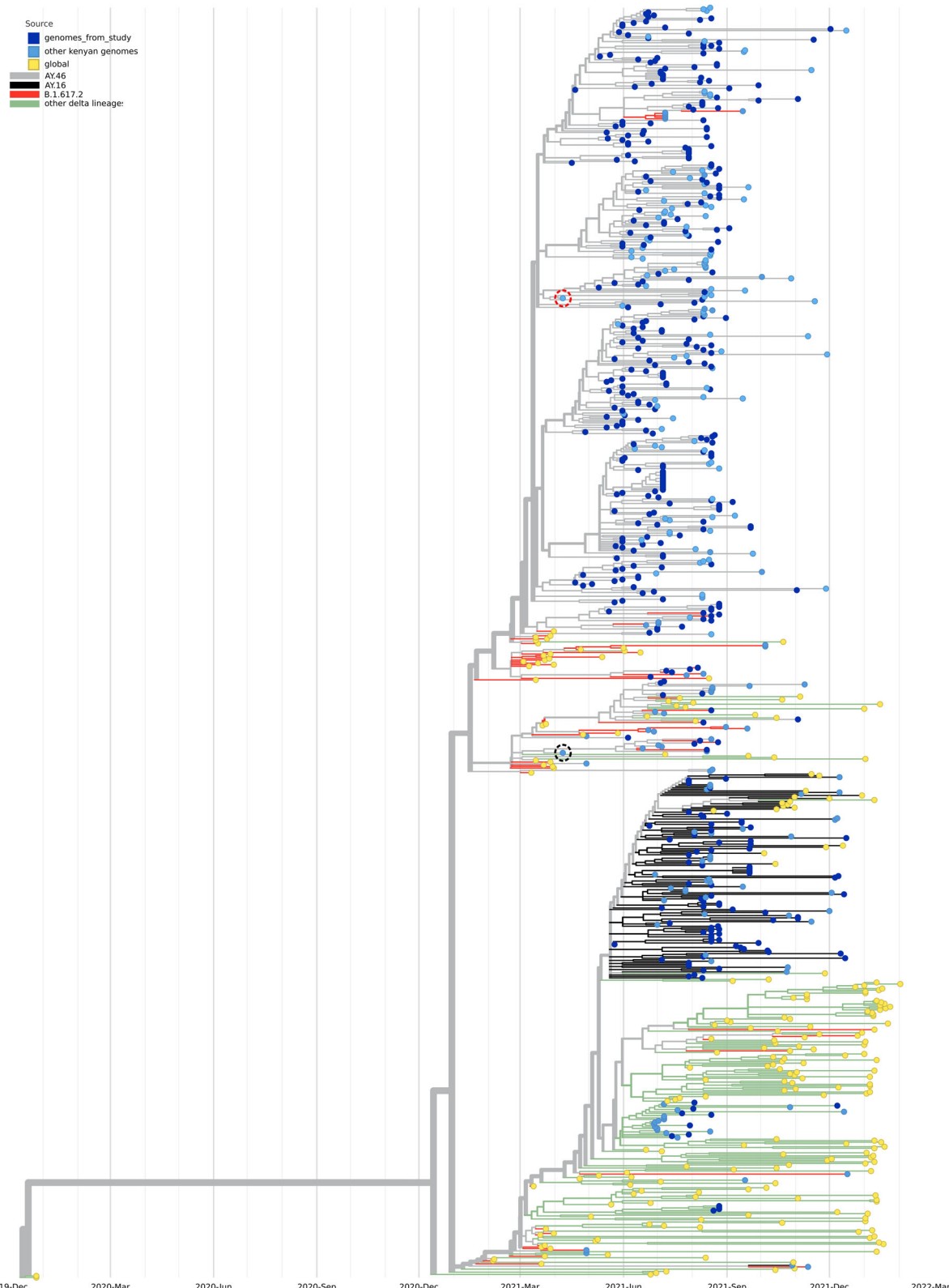

**Fig. 5 Phylogenetic tree of the B.1.617.2 lineage from our study samples and those from around the world.** The tree was constructed with 836 genomes, 605 from Kenya (396 from this study), and 231 sub-sampled from the globe (including 29 early B.1.617.2 lineages all from India (Red branches). Genomes from this study are shown in dark blue circular tips, while other Kenyan genomes are shown in light blue circular tips. Kenyan samples branched with AY.16 pango lineage (grey branches) and AY.46 lineage (Black branches). The AY.16 lineage was the majority, followed by the AY.46 lineage. Samples encircled in red (AY.16) and black (AY.61) represent the earliest reported Delta introductions in, both from Nairobi.

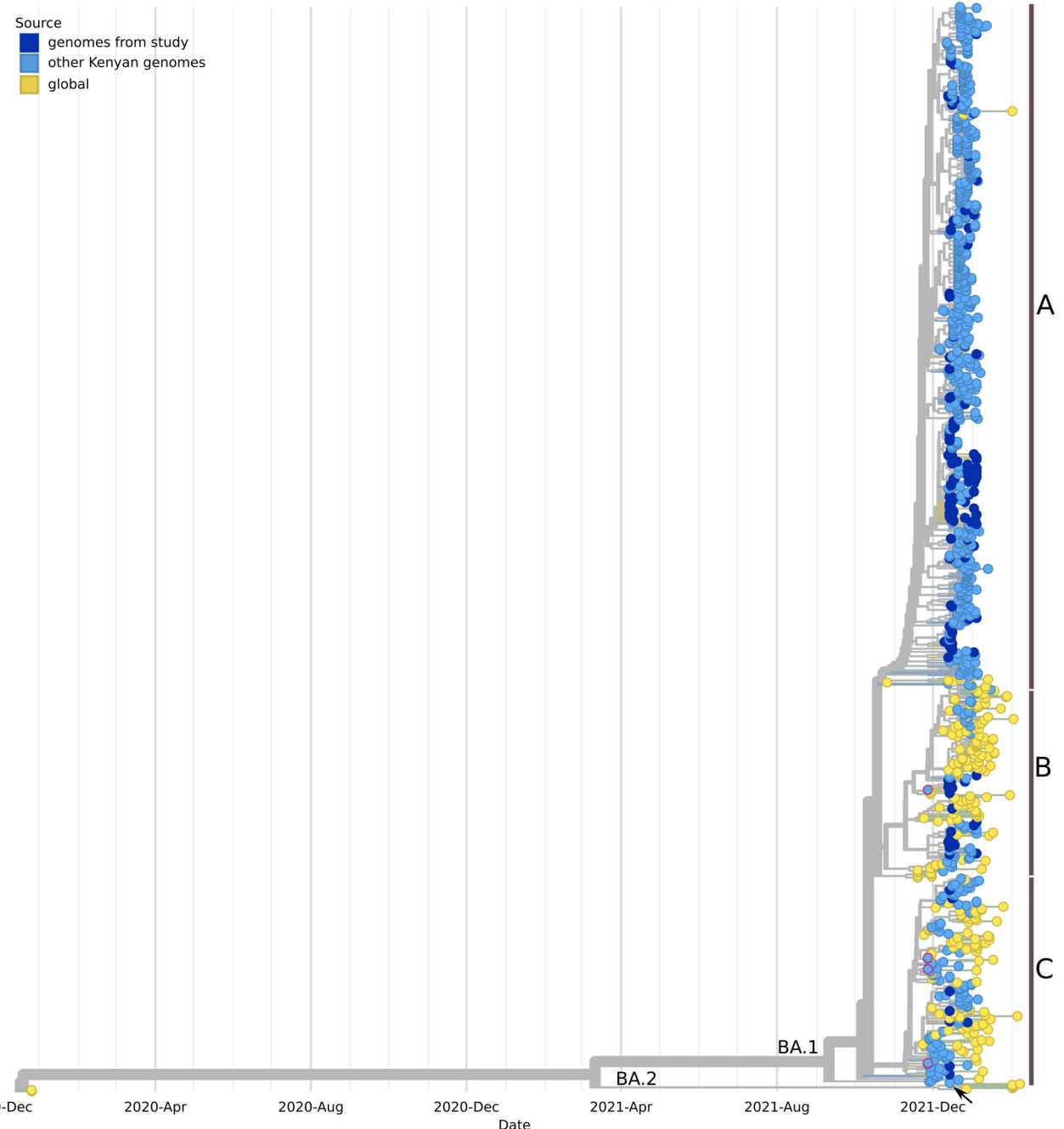

**Fig. 6 Phylogenetic tree of the omicron variant from our study samples and those from across the world.** The tree was constructed with 1041 genomes, 827 from Kenya (188 from our study) and 214 sub-sampled from the globe that included eight early omicron genomes mostly from South Africa. The tree was rooted against the Wuhan/WHO1/2019 and Wuhan/Hu-1/2019 SARS-CoV-2 reference genome genbank accession no.s LR757998 and MN908947, respectively. Genomes from the study are shown in dark blue circular tips while other Kenyan genomes are shown in light blue circular tips. The earliest Kenyan samples (27th November 2021) are shown in light blue tips with red borders. Other than one sample Kenya/ILRI_COVM01771/2021 - GISAID identifier EPI_ISL_9093518 that branched with lineage BA.2, all other samples branched with the BA.1 lineage. **A**, **B** and **C** represent lineage diversification within the BA.1 cluster. Most Kenyan samples branched within the A cluster.

## Discussion

In this study, we performed genomic surveillance of COVID-19 in the periods preceding and during the five waves that occurred in Kenya in order to understand the temporal dynamics of SARS-CoV-2 lineages. We show that each wave had a characteristic lineage (Fig. 1). Each wave was preceded by low infection rates, probably as variants competed through narrow transmission bottlenecks that selected the fittest variants[36], some of which eventually became the dominant variants. As shown in Fig. 2,

Kenyan samples branched into multiple lineages, illustrating multiple introduction events, and thereafter formed mono-phyletic clusters with notable intercluster divergence indicating ongoing local transmission. While 2020 was dominated by the European and local lineages of SARS-CoV-2, the subsequent year was dominated by VOCs that completely took over the COVID-19 scene.

The earliest SARS-CoV-2 samples sequenced at our laboratory were collected in May 2020. This was 2 months after the

confirmation of the first Kenyan case of COVID-19 on 12th March 2020[10]. Based on the genome sequences, the early SARS-CoV-2 was seeded by the European lineages (B.1 and B.1.1), and the A.25 Ugandan lineage (Fig. 1, panel A). The B.1 lineage dominated and had a countrywide distribution, having been identified in Nyanza region, Nairobi metropolitan area, Rift Valley, Western region, Central region and coastal Kenya, with Nairobi Metropolitan Area and the Western Region accounting for 38% and 25.6%, respectively (Supplementary Data 1 on Figshare). By August 2020, the B.1 lineage was the third most prevalent lineage globally, with 82,672 sequences deposited in the GISAID. This European lineage was first detected on 24th January 2020 and was most reported in North America and Europe. Its origin roughly corresponds to the Northern Italian outbreak[37]. In the Venn diagram (Fig. 1, panel B and Supplementary Table 1), the B.1 lineage was maintained across the first three COVID-19 waves, and dominated during wave one (55.8%) and wave two (56.3%). Its dominance however waned considerably by the third wave (1.8%). Other core lineages defining wave one were B.1.1 (European lineage that emerged in early February 2020), which occurred at a frequency of 22.5%, and A.25 (Ugandan lineage), which occurred at a frequency of 6.7%. Within the A.25 lineage, 6/8 samples were collected from trans-border truck drivers at Busia, a border town of Kenya and Uganda.

The B.1.549 lineage, mostly associated with Kenyan sequences and likely emerged from local transmission events was the second most prevalent lineage by wave two. The majority of samples in our dataset from this lineage were from the Kenyan coast. In a previous report of lineages detected across coastal Kenya counties between March 2020 and February 2021, the B.1.549 was found to occur at a frequency of 7.9%, and was the third most frequently observed lineage[16]. It is likely the lineage first emerged from coastal Kenya. The B.1.549 lineage was, however, not detected in the third wave, probably having been outcompeted to extinction by the more easily transmissible Alpha VOC. The last global report of the B.1.549 in GISAID was on January 29th, 2021 in Ohio, USA. Other local lineages that were present during wave two included B.1.530 (10.1%), B.1.596.1 (6.3%), N.8 (1.3%), B.1.428, B1.384 and the Ugandan lineage A.23. It is interesting to note that during the first and second waves, the local lineages persisted amidst the more dominant B.1 (Fig. 1A and Supplementary Table 1). The travel restrictions instituted early in these outbreaks may have allowed maintenance of local transmission events in the absence of external competing lineages.

The low infections between the waves allowed complacency in COVID-19 control practices, thus enabling the introduction and subsequent displacements of existing lineages by new variants. The only lineages that survived past wave two, albeit at low frequencies, were B.1, B.1.1, B.530 and A.23.1 (Fig. 1B). The dominance of the B.1 lineage in the previous two waves was eventually replaced by the Alpha VOC (B.1.1.7 lineage) that became the dominant lineage, accounting for 59.0% of all detected lineages in the early part of wave three (Fig. 1A). Our earliest sample with the B.1.1.7 lineage was on 1st February 2021 from two samples that came from Thika, Kiambu County, Kenya. Later that week, we detected the VOC in an outbreak that occurred in Nanyuki, Laikipia County and was traced to the British Army Training Unit in Kenya (BATUK). Between December 2020 and January 2021, the variant was rampant in the UK[38]. Phylogenetic analysis (Fig. 3, black arrows) estimated the BATUK outbreak to be the first major introduction of the Alpha variant into the country. The outbreak seemed to have been well contained, as there were no indications (with our data and Kenyan data deposited in GISAID) of out-branching from this outbreak cluster. Other Kenyan Alpha variant clusters appeared to have been introduced multiple times (Fig. 3, deep and light blue circular branch tips)

from independent sources. In less than three months after its detection, the Alpha variant had become the most dominant lineage. The Alpha VOC possesses several non-synonymous mutations of immunological importance[39] that are thought to confer increased transmissibility[40].

The Beta VOC (B.1.351 lineage) appears to have been introduced in Kilifi County, coastal Kenya in January 2021[16,41] and Fig. 4. Though not as highly transmissible as the Alpha and Delta variants, it has immune escape mutations[42,43], which could potentially compromise COVID-19 vaccines. Similar to the Alpha and Delta, the B.1.351 lineage emerged during the third wave, and in our dataset was the fourth most dominant lineage at 4.2% (Supplementary Data 1 on Figshare). Of all the beta variants deposited in GISAID from Kenya ($n = 184$: Date accessed 5 August 2021), 84.2% of them were from coastal Kenya, including Kilifi, Kwale and Mombasa counties. There is an over-representation of the B.1.351 lineage in the coastal region and being a popular tourist destination, it is possible that B.1.351 was introduced by visiting tourists to this region. For instance, the earliest reports of B.1.351 in Kenya were from South African travelers at the Coast in mid-December 2020[41]. Additionally, in a recent study, phylogeographic reconstruction tracking how the pandemic unfolded in Africa, the B.1.351 was shown to have been introduced into East Africa directly from southern Africa[20].

The Delta variant (B.1.617.2 lineage) was originally identified in India in October 2020[44,45]. It spread globally and in the process of adaptive evolution, delineated into sublineages, each with a distinct set of mutations, especially in the spike protein[46]. Currently there are a total of 238 Delta sublineages, with the AY.4 (UK/European lineage) being the most dominant globally[47]. In Kenya, the first report of Delta was on 5th May 2021, in Kisumu, the third largest city on the shores of Lake Victoria and was linked to travelers returning from India[48]. By the end of May 2021, Delta had become the dominant variant in Western Kenya[49]. Driven by its high transmissibility, estimated to be 60% higher than the Alpha variant[22], the variant soon extended its grip to the rest of the country. From the Kenyan genomes (Supplementary Fig. 1 and Fig. 1A), the Delta VOC was seeded in the third wave when infections rates with the Alpha VOC were still high. The variant slowly replaced the Alpha to become the dominant variant in the fourth wave. As shown in Fig. 1A and Supplementary Data 1 on Figshare, the fourth wave lasted the longest (July 2021 to December 2021). In our phylogenetic analysis, the oldest Kenyan samples were from Nairobi and had been collected on 8th April 2021. One was Delta sub lineage AY.16 and the other was AY.61 (Fig. 5, taxa encircled in red and black, respectively). The AY.16 lineage is described as a Kenyan lineage, and is commonly found in India 32% and Kenya 26%[50]. This lineage was the main cause of the Delta outbreak in Kenya, and accounted for 70% (283/404) of genomes from this study (Fig. 5, deep blue and light blue circled tips with grey branches). Its apparent cumulative prevalence, which is the ratio of AY.16 sequences collected since its identification in a particular location is the highest globally at 12%[51]. The secondary lineage behind the delta outbreak in Kenya was the AY.46 (Fig. 5 deep and light blue circled tips with black branches). This lineage was first identified on May 24th 2021 from a sample in Kisumu, and accounted for 17% (69/404) of the genomes collected in the current study.

In early November 2021, a hyper transmissible SARS-CoV-2 variant that was characterized by 37 mutations on the spike protein was identified in Southern Africa and thereafter designated Omicron VOC (B.1.1.529 lineage[52]). In Kenya, Omicron was first detected on 27th November 2021 from sequences collected in Nairobi, Kenya. At the time, the country was enjoying one of the lowest daily reported COVID-19 cases since the beginning of the pandemic (Supplementary Fig. 1), resulting in

complacency in COVID-19 control practices other than the mask mandates. Kenya, like the rest of the world was also entering into the Christmas and New Year festive seasons that are characterized by increased local and international travel as well as gatherings. These factors, coupled with hyper-transmissibility of Omicron facilitated its rapid spread across the country, and soon wave five commenced. In less than 30 days since the first reported Omicron genome in Kenya, the VOC had peaked to 3749 cases (25th December 2021), the highest since the start of the pandemic. Wave five was dominated by Omicron (95.4%) and a few Delta (4.7%). By the time of writing the manuscript, only three major sub-lineages of Omicron (BA.1, BA.2 and BA.3) had been reported globally[53]. From our data we only detected BA.1 and BA.2 (Supplementary Data 1 and 2 on Figshare). From our phylogenetic analysis, there was evidence of lineage diversification within the BA.1 cluster, into three major clusters A, B and C (Fig. 6). In Kenya, the omicron outbreak was largely caused by the BA.1.1 (Fig. 6, cluster A).

In addition to the VOCs, we also detected two variants of interest (VOI) The B.1.525 lineage (Eta), which has E484K, Q677H and F888L deletions, in addition to other deletion suite similar to B.1.1.7[37]. During wave three, this VOI was the third most prevalent at 7.4% and had a countrywide distribution, having been collected from Western Kenya (Busia, Kisumu, Migori and Nyamira Counties), Coastal Kenya (Mombasa and Kwale), Rift Valley (Nandi, Uasin Gishu Counties), Northern Kenya (Garissa), Eastern Kenya (Makueni) and Nairobi County. The other VOI was the A.23.1 lineage, an international lineage with variants of potential biological concern[37]. This variant contains a constellation of mutations, including E484K, that could reduce COVID-19 vaccine effectiveness[54,55]. The variant was dominant in the period between September and November 2020 in Uganda[54]. Most of the study samples with this lineage came from Busia, a major border town of Kenya and Uganda. It is likely that the lineage was seeded into Kenya from Uganda during cross-border trade and human movement.

Finally, as a limitation, it is important to point out that out of 1034 genomes evaluated, 32.6% were derived from Nyanza region, the rest being shared between Nairobi (18.1%), Rift Valley (18.0%), Western Kenya (13.3%), Central Kenya (12.7%) and the Coastal region (5.4%). This notwithstanding, our data shows similar trends in COVID-19 waves and the associated lineages as has been reported in the whole country[56].

## Conclusion

Five COVID-19 waves have occurred in Kenya since its introduction in March 2020, and by the time of writing, a sixth wave was predicated to occur in May or June 2022[57]. Each wave was fueled by different core sets of lineages. Wave one was seeded by imported lineages, mainly of European origin. The second wave had a mix of European and local lineages, the latter arising from local transmission and diversification events. The last three successive waves were dominated by imported VOCs that not only displaced lineages identified in waves one and two, but also edged each other out. Genomic surveillance will continue to play a critical role in generating SARS-CoV-2 lineage intelligence and especially cataloguing those associated with specific phenotypes, be they disease severity, increased transmissibility, diagnostic failure and/or vaccine breakthrough.

## Data availability

Assembled SARS-CoV-2 genomes from this study were uploaded to www.gisaid.org as FASTA files and their GISAID epi isl numbers are shown in Supplementary Data 1 on Figshare. Other source data for this study are provided as the supplementary dataset: https://doi.org/10.6084/m9.figshare.20085506.

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

## Acknowledgements

Kenya's MOH provided extraction and PCR kits. SARS-CoV-2 sequencing and personnel cost were funded by the Armed Forces Health Surveillance Division (AFHSD) and its Global Emerging Infections Surveillance and Research Branch (ProMIS P0095_21_KY, 2021). The funders had no role in data collection and analysis, decision to publish, or preparation of the manuscript. We thank all of the authors who have contributed genome data in GISAID (Supplementary Data 2 on Figshare).

## Author contributions

G.K. was in charge of genome sequencing and associated bioinformatics analysis, wrote the first draft, reviewed and edited the manuscript. J.N. was in charge of assays workflow, performed nucleic acids extractions, reviewed and edited the manuscript. E.O. was in charge of quality assurance/quality control, organized the qPCR assay workflow, analyzed and interpreted the qPCR data. F.S. assisted E.O. in qPCR quality assurance and quality control, performed qPCR assays, analyzed and interpreted the qPCR data. A.L. supervised the nucleic acid extractions workflow, assisted in nucleic acid extractions and together with E.O. determined samples that needed re-testing. E.M. was in charge of the cDNA library preparations, quality control and assisted in genome sequencing. B.A. received samples, made sample inventory, and assisted in nucleic acid extractions, and qPCR assays. R.L. together with B.A. received samples, made sample inventory, and assisted in nucleic acid extractions, and qPCR assays. R.G. was in charge of data curation and capture of associated metadata. C.M. together with E.O., organized the qPCR assay workflow, analyzed and interpreted the qPCR data. S.O., together with B.A. and R.L. received samples, made sample inventory, and assisted in nucleic acid extractions, and qPCR assays. G.A. was in charge of sample receipt, supervised sample retrieval and inventory. C.K. assisted in qPCR assay workflow, analysis and interpretation of the qPCR data, reviewed and edited the manuscript. B.M. ensured that all staff had appropriate personal protective equipment and that all biosafety cabinets were calibrated and safe to use, supervised all aspects of the assays, and revised and edited the manuscript. R.M.G. was in charge of samples accrual from Kenya Defence Forces' personnel, coordinated sample shipping to our testing laboratory, reviewed and edited the manuscript. J.W. conceived the study, reviewed and edited the manuscript, provided the resources, and obtained funding.

## Competing interests

The authors declare no competing interests.
