## [Peer Review File · Communications Medicine]

Reviewers' comments:

Reviewer #1 (Remarks to the Author):

Specific comments

Page 1, Line 28, include reference after the first sentence.

Page 1, Line 30, recast the phrase "As a result the citizens panicked"

Line 37-38, provide the appropriate reference to back the statement.

Page 2, Line 45, the statement needs to be backed by a suitable reference.

The authors should go through the background and ensure all statements stating or coating data are supported by an appropriate reference.

Methods

The statements in Lines 64 & 65 should be moved to the results section

Line 67, remove the phrase "In brief", not necessary

What do the authors mean by high coverage genomes in Line 82, this should be clearly defined.

The authors should also explain what they mean by context genomes in Line 91

The authors should please clearly state the criteria used in selection of sequences used in the 3 datasets generated for phylogenetic tree construction.

The authors made no mention of how they aligned their sequences and which tools and settings were used. The information regarding phylogenetic analysis was not provided, no mention of tree model or properties, please authors should include details of all bioinformatic methods/tools used not summary.

Results

The information provided in relation to sequenced samples too poor not sufficient. Authors are advised to include other information such as clinical condition/outcome, occupation, recent travel history, economic status e.t.c.

The structure of the results section also needs to be improved upon, they seem to be lumped up thereby capable of confusing readers, and also authors need to provide a separate table detailing the lineages identified in each wave and their sources/location. Also table 1 need to be expanded to include additional information as mentioned above.

Lines 129 to 175, are written in form of figure legend and not results, the authors are advised to move them into a separate section under the heading figure legends, and write out the data generated under the results section and not combine both.

The authors also need to support their finding with evidence showing spacial dispersal and phylogeographic movements of the various strains identified through the 3 waves under consideration in this study, a good example of such analysis can be seen in "Tegally et al. Nat Med. 2021 Mar;27(3):440-446." This will go a long way in highlighting the relationships between the origins and multiple introductions of the virus variants through the waves.

Reviewer #2 (Remarks to the Author):

Kamita et al provide an overview of the lineage composition and phylogenetic relationship of the viruses from the first three COVID-19 waves in Kenya following sequencing of 483 positive samples. Although there have been two genomic SARS-CoV-2 studies from Kenya(<https://www.nature.com/articles/s41467-021-25137-x> and

<https://www.medrxiv.org/content/10.1101/2021.07.01.21259583v1>), both focused on describing the early epidemic period and using coastal Kenya samples thus the situation in other Kenyan locations remain undocumented. The genomic composition of the third and fourth Kenya COVID-19 waves also remain undocumented. The work by Kamita et al confirm multiple SARS-CoV-2 lineage introductions and co-circulation during the first three Kenya waves observed up to June 2021. While wave 1&2 were dominated by the ancestral variant (D614G), wave 3 was dominated by VOCs, majorly Alpha. Although the authors did not analyze wave 4, by June 2021 delta variant was already in the country and rapidly moving towards predominance.

Comments for transmission to the authors

This reports attempts to fill in an important knowledge gap of SARS-CoV-2 Kenya using genomics. However, to draw out the full conclusions, some details are missing as highlighted below that are required to learn most from the data.

- 1) The study appears to use samples collected majorly from some parts of Western, Rift Valley and central regions of Kenya (although this is not clearly stated), how representative is your sample for the whole of Kenya to generalize the findings for the country? A table and/or map showing the sampling density across the country and time will be useful for the readers to understand your findings.
- 2) The criteria for selecting the samples sequenced needs clarification. The laboratory identified 4,109 positives during the study period but only 483 were genome sequenced. Other than Ct value <33, were there any additional criteria for a sample to be selected for sequencing? There is need to detail how many samples were considered at every stage and perhaps showing this in a sample flow gram.
- 3) Although work describing the original laboratory methods has been referenced as necessary, the methods in this work have minimal details available. For instance the whole genome sequencing procedure need to include reagent cat no. plus manufacturer name, quantities and concentrations used, number of reaction pools, thermocycling conditions, quality controls (e.g +ve/-ve Control) and , extent of multiplexing during sequencing.
- 4) Related to (3) above, the bioinformatic analysis needs details. For instance, the parameters used with ngs mapper v1.5, nature of curation done using the nextclade Web,, there are over 4M genomes on GISAID, how was the subsampling done to end-up with the few context genomes, exact version of Pangolin and Pangolearn used, how were multiple sequence alignments produced, details of the augur pipeline and parameters.
- 5) The background section needs references especially when describing the Kenya epidemic. There is now significant published literature on the Kenya /Africa epidemic including genomics work e.g(<https://pubmed.ncbi.nlm.nih.gov/34618602/>, <https://www.nature.com/articles/s41467-021-25137-x>, <https://pubmed.ncbi.nlm.nih.gov/34473191/>, <https://www.science.org/doi/10.1126/science.abj4336>, and <https://www.medrxiv.org/content/10.1101/2021.07.01.21259583v1>). This need to be discussed in the introduction to be clear on what knowledge gaps this study is attempting to fill and the context of the results here.

Minor observations

- It will be interesting to see some analysis on the estimated number of introductions for the different VOCs/VOI and a formal inference on the potential source
- Why is there no Beta specific tree albeit the being fewer in numbers?
- GISAID requires login to access the genomic data, do the authors plan to deposit the genomic data

in a fully public database e.g. GenBank?

- Line 26, not sure about the evidence of CIVET cats as intermediate hosts for SARS-CoV-2
- Line 14 and line 124, delta lineage is "B.1.617.2" NOT "B.1.167.2"
- Line 62, is Congo neighboring Kenya?
- Table 1: Are there participants who were repeatedly sampled? These will skew the participant demographics;
- It will be interesting to see the demographics of the tested individuals (positive vs negative) as this is a less biased sample compared to what is shown on Table 1 in understanding the underlying demographics
- In the methods, add information on who you defined wave (criteria) 1, 2, and 3 periods.
- Parts of line 110-111 are repeated in the discussion section line 179-180. This should be removed from the results section
- Line 206, I don't understand what you mean by "passed"
- It will be great if the authors can also highlight and discuss the limitations of this work.
- Other than genomes, do the authors plan to make the scripts and metadata used in their analysis public? Not seen any link for this.
- If the authors have acquired more genomic data during the review period, they should be encouraged to include it into this report to improve sample size

Reviewer #3 (Remarks to the Author):

This article is well written, original and in adequacy with the current problematic of the variants of SARS-CoV-2.

Analysis of variants associated with epidemic peaks in countries is important to understand the phenomenon of viral ecology shift in favor of more contagious variant which could with selective advantages.

Reviewer #4 (Remarks to the Author):

Comments on Kimita et al. "A genomics dissection of Kenya's COVID-19 waves: temporal lineage replacements and dominance of imported variants of concern"

The authors report an analysis of SARS-CoV-2 genomic sequences from a portion of the Kenyan Covid-19 epidemic. The work has the potential to be informative about viral movement into and evolution within Kenya. There are however a number issues with the manuscript and the analysis that could be improved.

1. Line 9 "Kenya's COVID-19 epidemic was slow to peak. It was seeded early in March 2020 and did not peak until late July 2020 (wave 1), mid-November 2020 (wave 2) and late March 2021 (wave 3)."

Line 244: "Three COVID-19 waves occurred in Kenya, and by the time of pressing, a 4th wave had emerged.

From the Our World in Data website (<https://ourworldindata.org/coronavirus>) there were 4 distinct waves in the OWID data with the most recent wave peaking in mid-August, nearly three months ago. For completeness all waves should be covered, including genomic sequencing across the 4th wave.

2. There are already a number of publications/preprints on the COVID-19 epidemic in Kenya which would be cited and discussed in the introduction as important background. Key papers covering Kenyan genomic surveillance that should be cited and the current results should be discussed in relation to what is already published about the epidemic in Kenya

Tracking the introduction and spread of SARS-CoV-2 in coastal Kenya.

Githinji et al. Nat Commun. 2021 Aug 10;12(1):4809. doi: 10.1038/s41467-021-25137-x.PMID: 34376689

Seroprevalence of Antibodies to SARS-CoV-2 among Health Care Workers in Kenya.

Etyang et al. Clin Infect Dis. 2021 Apr 24:ciab346. doi: 10.1093/cid/ciab346. Online ahead of print.PMID: 33893491

Genomic surveillance reveals the spread patterns of SARS-CoV-2 in coastal Kenya during the first two waves

Agoti, et al. medRxiv 2021.07.01.21259583; doi: <https://doi.org/10.1101/2021.07.01.21259583>

COVID-19 Transmission Dynamics Underlying Epidemic Waves in Kenya

Brand et al. medRxiv 2021.06.17.21259100; doi: <https://doi.org/10.1101/2021.06.17.21259100>

A year of genomic surveillance reveals how the SARS-CoV-2 pandemic unfolded in Africa. Wilkinson et al..Science. 2021 Oct 22;374(6566):423-431. doi: 10.1126/science.abj4336. Epub 2021 Sep 9.PMID: 34672751

4. Line 230: "Of all the beta variants deposited in GISAID from Kenya (n=184: Date accessed 5 August 2021), 84.2% of the B.1.351 lineages were from coastal Kenya, including Kilifi County, Kwale and Mombasa County. The overrepresentation of the B.1.351 lineage on the Kenyan Coast points to its possible introduction through the southern border with Tanzania and to the two tourists from South Africa."

The publications from the other groups in Kenya who generated these data (with much hard work) should be cited.

5. Line 9: "The data highlight the importance of genome surveillance in determining circulating variants to aid in public health interventions."

How are these sequence data used to aid in public health interventions? This should be better described.

6. Line 25: "zoonotic spillover event believed to have been from a progenitor bat coronavirus and civet cats as intermediates (Zhou et al., 2020)."

I don't know of any evidence of civet cats as an intermediate for SARS-CoV-2 and there is nothing in the Zhou et al. 2020 reference to support this statement. Perhaps the authors are confusing this with SARS-CoV.

7. Figure 1. Odd X axis, the unit changes, some times every day, sometimes very 2 or 3 days. The authors would be better off plotting lineages detected by month or by week at best, it would also be

easier to see the lineage changes. The lineages are almost impossible to discern, poor choice of color (e.g. there are 6 nearly identical dark blue/blacks used) and the X axis unit is too fine. Also case numbers across the country should be plotted as a reference to show when the three waves occurred.

8. Figure 2: Venn plot showing unique and shared lineages across the three COVID-19 waves. Without quantitative visualization, this figure is fairly non-informative and could be dropped.

9. Figure 3. "Time-scaled phylogenetic tree of Kenyan samples against global isolates. The tree was constructed with 112 genomes sampled from GISAID and 323 genomes from this study. Thin lines represent context global samples, while thick lines represent Kenyan samples. The different colors on circular tips of branches represent the Pango lineages."

How were these 112 GISAID genomes selected? Not clear which nodes are from this study. Not clear what question is being asked with this analysis.

10. Figure 4. Not clear what question is being asked with this analysis. How were global B.1.1.7 sequences selected? Do we need to distinguish every global country? It is hard to discern the Kenyan nodes with similar red, dark oranges used for Kenya, Denmark, Libya, South Sudan.

11. Line 158 "Wuhan/WHO1/2019 reference" Not clear what this is. the authors should specify a Genbank or GISAID accession number.

12. Figure 5. "Phylogenetic tree of the B.1.617.2 lineage from our study samples and those from across Africa. The tree was constructed with 893 genomes, including those from Kenya (n=33), those from other parts of Africa (n=812) and early B.1.617.2 lineages (n=46) traceable to India. Kenyan samples are shown as circular red branch tips. The red stars show the earliest delta variant introduction in late April 2021 from Nairobi samples, while samples from Kisumu (the county that had the first major delta variant outbreak) are contained in clades represented by blue and purple stars."

13. Why were only genomes from African countries included global data used for Figure 5?. There is much tourist and European traffic into Kenya and a substantial Indian population so it is equally likely that B.1.617.2 variants entered from many parts of the world. What is the question being asked with this analysis?

14. Line 102 and Table 1 demographics. Alone, the demographic data of the sequenced samples are not very meaningful without the sample parameters for all COVID-19 cases and for all of Kenya for comparison. Is the median age of 33 different from the median age of all diagnosed COVID-19 cases and from the median age of the total population? The data from all cases is needed if the authors want to argue that there is no bias in the sequenced samples. I suspect the pattern will be similar to Kenya from the region sampled. But again, what is the question being asked with this analysis?

15. Line 109 "Each wave was preceded by low infection rates, probably as variants competed through narrow transmission bottlenecks that selected the fittest variants, some of them to eventually become the dominant variants in succeeding waves (Lythgoe et al., 2021)."

This is speculation and should not be in results section. There could be a lot of reasons for these

patterns and the authors present this as a fact, which can be misleading. Also I doubt that the selection occurred in Kenya, the VOC B.1.1.7 and B.1.251 entered from UK or South Africa already fit.

16. Line 21: "has literally been the 2020/21 blockbuster virus"

Considering the large number of deaths and the long-term sequelae of this infection, referring to the virus as a "blockbuster" (which has bestseller book and movie connotations) seems glib and an inappropriate word choice. I would change this

Responses to the editor's and reviewers comments

Reviewer #1 (Remarks to the Author):

Specific comments

Page 1, Line 28, include reference after the first sentence.

The reviewer may have missed the reference. See lines 43 in the MS with tracked changes

Page 1, Line 30, recast the phrase "As a result the citizens panicked"

The statement has now been changed to "To try to curtail the spread, the government instituted a series of countermeasures that included border closures, mandatory quarantine on returning travelers, night curfews, ban on gatherings, and mandatory mask use while in public spaces (Ministry of Health Kenya, 2020b)". See lines 43-46 in the MS with tracked changes.

Line 37-38, provide the appropriate reference to back the statement.

A reference has now been added. See reference 16, lines 55 in the MS with tracked changes.

Page 2, Line 45, the statement needs to be backed by a suitable reference.

We thank the reviewer for catching this omission. The appropriate references have now been added. See lines 65 and 66 in the MS with tracked changes.

The authors should go through the background and ensure all statements stating or coating data are supported by an appropriate reference.

All appropriate references in the background section have been included, and also added to the list of references.

Methods

The statements in Lines 64 & 65 should be moved to the results section

We think that this statement is appropriate in this section as it explains how samples were selected for whole genome sequencing.

Line 67, remove the phrase "In brief", not necessary

Done. See line 90 in the MS with tracked changes.

What do the authors mean by high coverage genomes in Line 82, this should be clearly defined.

This whole section has now been rewritten. See section titled “Global data acquisition”, lines 128 to 138 in the MS with tracked changes.

The authors should also explain what they mean by context genomes in Line 91

The word “context genome” is used synonymously as “comparator genome”.

The authors should please clearly state the criteria used in selection of sequences used in the 3 datasets generated for phylogenetic tree construction.

We have now expounded the sub-sampling criteria, under the new section “Global data acquisition”. See lines 128 to 138 in the MS with tracked changes.

The authors made no mention of how they aligned their sequences and which tools and settings were used. The information regarding phylogenetic analysis was not provided, no mention of tree model or properties, please authors should include details of all bioinformatic methods/tools used not summary.

We thank the reviewer for catching this omission. Details requested by the reviewer are now provided. See lines 140 to 155 and lines 156 to 161 in the MS with tracked changes.

Results

The information provided in relation to sequenced samples too poor not sufficient. Authors are advised to include other information such as clinical condition/outcome, occupation, recent travel history, economic status e.t.c.

Additional epidemiological data available to us are now provided in Table 1 and Supplementary Table 1. See lines 166 to 172 in the MS with tracked change. Unfortunately, being a testing Lab, we did not receive enough epidemiological information.

The structure of the results section also needs to be improved upon, they seem to be lumped up thereby capable of confusing readers, and also authors need to provide a separate table detailing the lineages identified in each wave and their sources/location. Also table 1 need to be expanded to include additional information as mentioned above.

The results’ section has been restructured with subtitles to make reading easier. Table detailing lineages identified in each wave, their sources and location are now provided as Supplementary Table 1. Also, as stated above, we did not receive enough epidemiological information.

Lines 129 to 175, are written in form of figure legend and not results, the authors are advised to move them into a separate section under the heading figure legends, and write out the data generated under the results section and not combine both.

Am not sure I understand what the reviewer is unhappy about. Figure legends are provided in the following their first mention. Anyhow, we have tried to make the legends and results more distinct.

The authors also need to support their finding with evidence showing spacial dispersal and phylogeographic movements of the various strains identified through the 3 waves under consideration in this study, a good example of such analysis can be seen in “Tegally et al. Nat Med. 2021 Mar;27(3):440-446.” This will go a long way in highlighting the relationships between the origins and multiple introductions of the virus variants through the waves.

Unfortunately, most our samples did not come with enough epidemiological information to allow the analysis proposed by the reviewer. We have nevertheless speculated on sources, especially for VOCs. See lines 277 to 359 in the manuscript with tracked changes.

Reviewer #2 (Remarks to the Author):

1) The study appears to use samples collected majorly from some parts of Western, Rift Valley and central regions of Kenya (although this is not clearly stated), how representative is your sample for the whole of Kenya to generalize the findings for the country? A table and/or map showing the sampling density across the country and time will be useful for the readers to understand your findings.

The reviewer is correct that our study genomes are not representative of the whole country. We now state the number of samples received from each region (see lines 171 to 172). We have also included Supplementary Table 1 that shows place and date of sample collection. Lastly, we acknowledge the data skew as a limitation (see lines 465 to 468 in the manuscript with tracked changes).

2) The criteria for selecting the samples sequenced needs clarification. The laboratory identified 4,109 positives during the study period but only 483 were genome sequenced. Other than Ct value <33, were there any additional criteria for a sample to be selected for sequencing? There is need to detail how many samples were considered at every stage and perhaps showing this in a sample flow gram.

We have now provided additional information on criteria for sample selection. We say “Of the 1089 COVID-19 nasal specimens that passed the threshold for whole genome sequencing (Cts <33), 45 were dropped because they did not pass the threshold required for assigning Pango lineages. Ten additional samples were dropped because they lacked date of collection. The remaining 1034 genomes collected between May 2020 and January 2022 were used to monitor the evolution of SARS-CoV-2 lineages across the five COVID-19 waves. See lines 124 to 127 in the manuscript with tracked changes.

3) Although work describing the original laboratory methods has been referenced as necessary, the methods in this work have minimal details available. For instance the whole genome sequencing procedure need to include reagent cat no. plus manufacturer name, quantities and concentrations used, number of reaction pools, thermocycling conditions, quality controls (e.g +ve/-ve Control) and , extent of multiplexing during sequencing.

We thank the reviewer for catching these discrepancies. The Methods section has been elaborated to include more details than previously provided. See lines 90 to 103 in the manuscript with tracked changes.

4) Related to (3) above, the bioinformatic analysis needs details. For instance, the parameters used with ngs mapper v1.5, nature of curation done using the nextclade Web,, there are over 4M genomes on GISAID, how was the subsampling done to end-up with the few context genomes, exact version of Pangolin and Pangolearn used, how were multiple sequence alignments produced, details of the augur pipeline and parameters.

We again thank the reviewer for catching these discrepancies. More details are now provided in the Methods section, to include more details than previously provided. See lines 129 to 161 in the manuscript with tracked changes.

5) The background section needs references especially when describing the Kenya epidemic. There is now significant published literature on the Kenya /Africa epidemic including genomics work e.g(<https://pubmed.ncbi.nlm.nih.gov/34618602/>, <https://www.nature.com/articles/s41467-021-25137-x>, <https://pubmed.ncbi.nlm.nih.gov/34473191/>, <https://www.science.org/doi/10.1126/science.abj4336>, and <https://www.medrxiv.org/content/10.1101/2021.07.01.21259583v1>). This need to be discussed in the introduction to be clear on what knowledge gaps this study is attempting to fill and the context of the results here.

We have now expanded the references in the background section and included the relevant literature, as suggested by the reviewer.

Minor observations

- It will be interesting to see some analysis on the estimated number of introductions for the different VOCs/VOI and a formal inference on the potential source

Unfortunately, most our samples did not come with enough epidemiological information to allow the analysis proposed by the reviewer. We have nevertheless speculated on sources, especially for VOCs. See lines 277 to 359 in the manuscript with tracked changes.

-Why is there no Beta specific tree albeit the being fewer in numbers?

We have now included a Beta VOC phylogenetic tree. See Figure 4, described in lines 310 to 324 in the manuscript with tracked changes.

-GISAID requires login to access the genomic data, do the authors plan to deposit the genomic data in a fully public database e.g. GenBank?

Yes, once our manuscript is published, we deposit the genomic data in a freely accessible public database.

- Line 26, not sure about the evidence of CIVET cats as intermediate hosts for SARS-CoV-2

This section has been rewritten. We say that “The origin of SARS-CoV-2 is controversial (Andersen et al., 2020) but from genetic studies, the closest relatives are bat coronaviruses believed to have been transmitted to humans as a result of a zoonotic spillover event (Li et al., 2020; Wan et al., 2020; Wei et al., 2020; Zhou et al., 2020). It remains to be proven whether the bat facilitated both the evolution of SARS-CoV-2 and also its transmission to humans (Banerjee et al., 2021). See lines 32 to 41 in the manuscript with tracked changes.

- Line 14 and line 124, delta lineage is “B.1.617.2” NOT “B.1.167.2”.

Thank you. We corrected the errors. See lines 16 and 208 in the manuscript with tracked changes.

-Line 62, is Congo neighboring Kenya?

This has been corrected (see lines 84 in the manuscript with tracked changes).

- Table 1: Are there participants who were repeatedly sampled? These will skew the participant demographics;

No, the data does not include participants with repeat sampling

- It will be interesting to see the demographics of the tested individuals (positive vs negative) as this is a less biased sample compared to what is shown on Table 1 in understanding the underlying demographics

We agree with the reviewer that a comparison of demographics between individuals who tested positive Vs negative would be interesting. Our study was however on individuals who provided useable genome sequences and not even on those who tested positive.

- In the methods, add information on who you defined wave (criteria) 1, 2, and 3 periods.

Thanks for pointing this out. We have added a sentence on how the waves were defined. We say, “COVID-19 waves were defined by observing increase and decrease of positive samples over time and in general corresponded to the waves observed in the whole country (Extended data 1)”. See lines 178 to 180 in the manuscript with tracked changes.

-Parts of line 110-111 are repeated in the discussion section line 179-180. This should be removed from the results section

The statement has been removed

- Line 206, I don't understand what you mean by "passed"

Again, thanks for catching this typo. Now corrected to "past". See line 399 in the manuscript with tracked changes.

-It will be great if the authors can also highlight and discuss the limitations of this work.

A paragraph highlighting limitations of the study has now been added. See lines 465 to 468 in the manuscript with tracked changes.

- Other than genomes, do the authors plan to make the scripts and metadata used in their analysis public? Not seen any link for this.

We thank the reviewer for this suggestion. We had not planned on doing this, but come to think of it, we should. We will eventually make them accessible to the public.

-If the authors have acquired more genomic data during the review period, they should be encouraged to include it into this report to improve sample size

Great suggestion. We have done just that. We now include data collected up to January 2022

Reviewer #3 (Remarks to the Author):

This article is well written, original and in adequacy with the current problematic of the variants of SARS-CoV-2.

Analysis of variants associated with epidemic peaks in countries is important to understand the phenomenon of viral ecology shift in favor of more contagious variant which could with selective advantages.

Reviewer #4 (Remarks to the Author):

Comments on Kimita et al. "A genomics dissection of Kenya's COVID-19 waves: temporal lineage replacements and dominance of imported variants of concern"

The authors report an analysis of SARS-CoV-2 genomic sequences from a portion of the Kenyan Covid-19 epidemic. The work has the potential to be informative about viral movement into and evolution within Kenya. There are however a number issues with the manuscript and the analysis that could be improved.

1. Line 9 "Kenya's COVID-19 epidemic was slow to peak. It was seeded early in March 2020 and did not peak until late July 2020 (wave 1), mid-November 2020 (wave 2) and late March 2021 (wave 3)."

This has been corrected. See lines 10-18 in the manuscript with tracked changes.

Line 244: "Three COVID-19 waves occurred in Kenya, and by the time of pressing, a 4th wave had emerged.

From the Our World in Data website (<https://ourworldindata.org/coronavirus>) there were 4 distinct waves in the OWID data with the most recent wave peaking in mid-August, nearly three months ago. For completeness all waves should be covered, including genomic sequencing across the 4th wave.

Great suggestion. We have done just that. We now include data collected up to January 2022, that includes all the five waves.

2. There are already a number of publications/preprints on the COVID-19 epidemic in Kenya which would be cited and discussed in the introduction as important background. Key papers covering Kenyan genomic surveillance that should be cited and the current results should be discussed in relation to what is already published about the epidemic in Kenya.

Tracking the introduction and spread of SARS-CoV-2 in coastal Kenya.

Githinji et al. Nat Commun. 2021 Aug 10;12(1):4809. doi: 10.1038/s41467-021-25137-x.PMID: 34376689

Seroprevalence of Antibodies to SARS-CoV-2 among Health Care Workers in Kenya.

Etyang et al. Clin Infect Dis. 2021 Apr 24;ciab346. doi: 10.1093/cid/ciab346. Online ahead of print.PMID: 33893491

Genomic surveillance reveals the spread patterns of SARS-CoV-2 in coastal Kenya during the first two waves

Agoti, et al. medRxiv 2021.07.01.21259583; doi: <https://doi.org/10.1101/2021.07.01.21259583>

COVID-19 Transmission Dynamics Underlying Epidemic Waves in Kenya

Brand et al. medRxiv 2021.06.17.21259100; doi: <https://doi.org/10.1101/2021.06.17.21259100>

A year of genomic surveillance reveals how the SARS-CoV-2 pandemic unfolded in Africa. Wilkinson et al..Science. 2021 Oct 22;374(6566):423-431. doi: 10.1126/science.abj4336. Epub 2021 Sep 9.PMID: 34672751

These suggestions were as also made by reviewer 2, comment #5. We have now expanded the references in the background section and included the relevant literature, as suggested by the reviewers.

4. Line 230: "Of all the beta variants deposited in GISAID from Kenya (n=184: Date accessed 5 August 2021), 84.2% of the B.1.351 lineages were from coastal Kenya, including Kilifi County, Kwale and Mombasa County. The overrepresentation of the B.1.351 lineage on the Kenyan Coast points to its possible introduction through the southern border with Tanzania and to the two tourists from South

Africa."

The publications from the other groups in Kenya who generated these data (with much hard work) should be cited.

We cannot find a specific paper on Beta variant, but we have cited the Policy brief reported by the Wellcome Trust group in Kilifi, Kenya (KEMRI, 2021). We say “The Beta VOC (B.1.351 lineage) appears to have been introduced in Kilifi County, coastal Kenya in January 2021 (KEMRI, 2021)”. See line 415 to 422 the manuscript with tracked changes.

5. Line 9: "The data highlight the importance of genome surveillance in determining circulating variants to aid in public health interventions."

How are these sequence data used to aid in public health interventions? This should be better described.

We have rephrased the sentence to say “The data highlight the importance of genome surveillance in determining circulating variants to aid interpretation of phenotypes such as transmissibility, virulence and/or resistance to therapeutics/vaccines”. See lines 26-26 in the manuscript with tracked changes.

6. Line 25: "zoonotic spillover event believed to have been from a progenitor bat coronavirus and civet cats as intermediates (Zhou et al., 2020)."

I don't know of any evidence of civet cats as an intermediate for SARS-CoV-2 and there is nothing in the Zhou et al. 2020 reference to support this statement. Perhaps the authors are confusing this with SARS-CoV.

We thank the reviewer for catching confusion. This section has been rewritten. “The origin of SARS-CoV-2 is controversial (Andersen et al., 2020) but from genetic studies, the closest relatives are bat coronaviruses believed to have been transmitted to humans as a result of a zoonotic spillover event (Li et al., 2020; Wan et al., 2020; Wei et al., 2020; Zhou et al., 2020). It remains to be proven whether the bat facilitated both the evolution of SARS-CoV-2 and also its transmission to humans (Banerjee et al., 2021). See lines 32 to 41 in the manuscript with tracked changes.

7. Figure 1. Odd X axis, the unit changes, some times every day, sometimes very 2 or 3 days. The authors would be better off plotting lineages detected by month or by week at best, it would also be easier to see the lineage changes. The lineages are almost impossible to discern, poor choice of color (e.g. there are 6 nearly identical dark blue/blacks used) and the X axis unit is too fine. Also case numbers across the country should be plotted as a reference to show when the three waves occurred.

We agree with the reviewer that the x-axis scale was too fine. We have now collapsed the x-axis into a month-year system and also collapsed the lineages based on their virulence i.e. VOC/VOI vs the other non-VOC/VOI. Case numbers across the country have now been plotted and are available as

“Extended data 1”.

8. Figure 2: Venn plot showing unique and shared lineages across the three COVID-19 waves. Without quantitative visualization, this figure is fairly non-informative and could be dropped.

We humbly disagree with the reviewer. Although the Venn plot is not quantitative, it provides a good visual summary of lineage relationships across the waves.

9. Figure 3. "Time-scaled phylogenetic tree of Kenyan samples against global isolates. The tree was constructed with 112 genomes sampled from GISAID and 323 genomes from this study. Thin lines represent context global samples, while thick lines represent Kenyan samples. The different colors on circular tips of branches represent the Pango lineages."

How were these 112 GISAID genomes selected? Not clear which nodes are from this study. Not clear what question is being asked with this analysis.

The genome selection criteria was also requested by reviewer 3, comment #4. We have now provided more details in genome selection. See lines 128 to 138 in the manuscript with tracked changes.

The Time-scaled phylogenetic tree of Kenyan genomes against selected global sequences is now Figure 2. The nodes from study samples are shown in deep blue circular tips, other Kenyan genomes in light blue and global genomes in Yellow circular tips.

As stated in the text, the figure shows the phylogenetic placement of the study genomes in a global context. In the new figure, we place more emphasis on VOIs or VOCs, and color these branches appropriately.

10. Figure 4. Not clear what question is being asked with this analysis. How were global B.1.1.7 sequences selected? Do we need to distinguish every global country? It is hard to discern the Kenyan nodes with similar red, dark oranges used for Kenya, Denmark, Libya, South Sudan.

This tree, now Figure 4 has been re-done. The analysis sought to identify how the alpha variants in Kenya relate phylogenetically to those sub-sampled globally. We also track lineages introductions and subsequent diversification. The study samples are now shown in deep blue circular tips, other Kenyan genomes in light blue and global genomes in yellow circular tips.

11. Line 158 "Wuhan/WHO1/2019 reference" Not clear what this is. the authors should specify a Genbank or GISAID accession number.

Thank you for noting this omission. Genbank accession number has been added. See lines 303 in the manuscript with tracked changes. We have also added Genbank/GISAID accession numbers for the SAR-CoV-2 references we used to root all the trees.

12. Figure 5. "Phylogenetic tree of the B.1.617.2 lineage from our study samples and those from across Africa. The tree was constructed with 893 genomes, including those from Kenya (n=33), those from other parts of Africa (n=812) and early B.1.617.2 lineages (n=46) traceable to India. Kenyan samples are shown as circular red branch tips. The red stars show the earliest delta variant introduction in late April 2021 from Nairobi samples, while samples from Kisumu (the county that had the first major delta variant outbreak) are contained in clades represented by blue and purple stars."

13. Why were only genomes from African countries included global data used for Figure 5?. There is much tourist and European traffic into Kenya and a substantial Indian population so it is equally likely that B.1.617.2 variants entered from many parts of the world. What is the question being asked with this analysis?

The reviewer raises a good point. We have now changed the approach and used a global sub-sample rather than a regional sub-sampling, owing to a solid argument of high traffic between Kenya and the world. The analysis sought to identify how the delta variants in Kenyan relate phylogenetically to those sub-sampled globally. We also track first introductions and subsequent diversification, especially for clusters such as the AY.46 and AY.16 that were driving the outbreak in Kenya.

14. Line 102 and Table 1 demographics. Alone, the demographic data of the sequenced samples are not very meaningful without the sample parameters for all COVID-19 cases and for all of Kenya for comparison. Is the median age of 33 different from the median age of all diagnosed COVID-19 cases and from the median age of the total population? The data from all cases is needed if the authors want to argue that there is no bias in the sequenced samples. I suspect the pattern will be similar to Kenya from the region sampled. But again, what is the question being asked with this analysis?

We humbly disagree with the reviewer's view on Table 1. First, the Table shows data on individuals who contributed genomes, not to be confused with those who had SARS-CoV-2 by PCR. There is a bias on sequenced samples in that only those with enough viremia (Ct <33) were sequenced. In the section titled "Sample acquisition" we say: "Of the 63,542 tested, 8.59.0% (n=5,3754) were positive for SARS-CoV-2 at varying cycle thresholds (Ct). Of these positive samples, 1089 with Cts <33 were selected for whole genome sequencing". See lines 91 to 92 in the manuscript with tracked changes.

15. Line 109 "Each wave was preceded by low infection rates, probably as variants competed through narrow transmission bottlenecks that selected the fittest variants, some of them to eventually become the dominant variants in succeeding waves (Lythgoe et al., 2021)."

This is speculation and should not be in results section. There could be a lot of reasons for these patterns and the authors present this as a fact, which can be misleading. Also I doubt that the selection occurred in Kenya, the VOC B.1.1.7 and B.1.251 entered from UK or South Africa already fit.

The statement has now been removed.

16. Line 21: "has literally been the 2020/21 blockbuster virus"

Considering the large number of deaths and the long-term sequelae of this infection, referring to the virus as a "blockbuster" (which has bestseller book and movie connotations) seems glib and an inappropriate word choice. I would change this

We agree with the reviewer. The sentence has been rephrased to read ... "Severe acute respiratory syndrome coronavirus 2 (SARS-CoV-2) has impacted public health, social, political and economic spheres of life worldwide since its emergence in Wuhan, China and subsequent spread to the rest of the world.". See lines 29 to 31 in the manuscript with tracked changes.

REVIEWERS' COMMENTS:

Reviewer #1 (Remarks to the Author):

The authors have addressed all the concerns raised, the manuscript can now be accepted for publication

Reviewer #2 (Remarks to the Author):

The authors are to be commended for providing details on the study method that were missing in the previous draft, as well as adding additional genome sequence data (wave 4 and 5) to the revised manuscript. The revised draft provides a more comprehensive overview of genomic epidemiology of the SARS-CoV-2 virus in Kenya over the five waves thus far observed, while appropriately acknowledging the study limitations. It is especially informative to see the lineage and phylogeny of the 5 waves of infections experienced in Kenya.

The authors should now carefully review the manuscript and edit throughout to achieve high accuracy, remove repetitive words/sections, spelling mistakes, and informal writing, for example.

The opening statement of the abstract (“...was slow to peak”) is only relevant to the first wave. “...late- mid August 2020”, should be “mid-late August 2020”

In the abstract Wave 1 and 2 have lineage proportions while wave 3-5 have numbers, why? The proportions given in line 17 become confusing.

Line 17 had double full-stop.

Line 24.... 504+,..... 6.2+, also in line 39 322+ and 5k+

Line 26, I am not sure why the authors delve into the controversy of SARS-CoV-2 origin. I don't see this a relevant to the molecular epidemiology in Kenya.

Line 33 requires a reference.

Line 48, I am not sure the waves were turning over because of counter measures alone. Susceptible population may have reduced as well, or may be local weather patterns etc

The discussion has some of the results restated e.g line 295. This should be avoided for brevity.

Line 328, Delta variant is B.1.617.2 not B.1.167.2

Reviewer #4 (Remarks to the Author):

Comments on the revision of "A genomics dissection of Kenya's COVID-19 waves: temporal lineage replacements and dominance of imported variants of concern" by Dr Waitumbi and colleagues [Paper # COMMSMED-21-0485A]

The authors have addressed most of my concerns except point 8 and point 14. Both analyses add clutter and little new information to an already too-long paper. The editor should make a decision about the inclusion of these two figures.

8. Figure 2: Venn plot showing unique and shared lineages across the three COVID-19 waves. Without quantitative visualization, this figure is fairly non-informative and could be dropped.

We humbly disagree with the reviewer. Although the Venn plot is not quantitative, it provides a good visual summary of lineage relationships across the waves.

My comments: Figure 2 should be dropped, it is non-informative in an already too-long paper.

14. Line 102 and Table 1 demographics. Alone, the demographic data of the sequenced samples are not very meaningful without the sample parameters for all COVID-19 cases and for all of Kenya for comparison. Is the median age of 33 different from the median age of all diagnosed COVID-19 cases and from the median age of the total population? The data from all cases is needed if the authors want to argue that there is no bias in the sequenced samples. I suspect the pattern will be similar to Kenya from the region sampled. But again, what is the question being asked with this analysis?

We humbly disagree with the reviewer's view on Table 1. First, the Table shows data on individuals who contributed genomes, not to be confused with those who had SARS-CoV-2 by PCR. There is a bias on sequenced samples in that only those with enough viremia (Ct <33) were sequenced. In the section titled "Sample acquisition" we say: "Of the 63,542 tested, 8.59.0% (n=5,3754) were positive for SARS-CoV-2 at varying cycle thresholds (Ct). Of these positive samples, 1089 with Cts <33 were selected for whole genome sequencing". See lines 91 to 92 in the manuscript with tracked changes.

My comments: Such an analysis risks overinterpreting the available data and I continue to recommend that this table and analysis be dropped.

Responses to the editor's and reviewers comments

Reviewer #1 (Remarks to the Author):

The authors have addressed all the concerns raised, the manuscript can now be accepted for publication.

Reviewer #2 Comments

The opening statement of the abstract (“...was slow to peak”) is only relevant to the first wave. “...late- mid August 2020”, should be “mid-late August 2020”

Thanks for catching this. We have rechecked Kenya's COVID-19 data and adjusted the statement to read ... “Kenya’s COVID-19 epidemic was seeded early in March 2020 and did not peak until early August 2020. See lines 11 in the MS with tracked changes

In the abstract Wave 1 and 2 have lineage proportions while wave 3-5 have numbers, why? The proportions given in line 17 become confusing.

We have re-written the statement to include the exact numbers of the lineage during those waves, to now read... “The B.1 lineage continued to expand and remained dominant, accounting for 60% (72/120) and 57% (45/79) in waves 1 and 2 respectively”. In this statement, we want to show the reader that the B.1 lineage was the dominant lineage during waves 1 and 2. See lines 17 in the MS with tracked changes

We agree with the reviewer on the confusion of the indicated proportions. We have now given the proportions of lineages per each wave. The sentence now reads.... “Waves three, four and five respectively were dominated by VOCs that were distributed as follows: Alpha 58.5% (166/285), Delta 92.4% (327/354), Omicron 95.4% (188/197) and Beta at 4.2% (12/284) during wave 3 and 0.3% (1/354) for wave 4. See lines 18 in the MS with tracked changes

Line 17 had double full-stop.

The double full removed.

Line 24.... 504+,..... 6.2+, also in line 39 322+ and 5k+

The numbers have now been re-written and updated. See lines 28 in the MS with tracked changes

Line 26, I am not sure why the authors delve into the controversy of SARS-CoV-2 origin. I don't see this a relevant to the molecular epidemiology in Kenya.

We believe this is very important background information for SARS-CoV-2, especially because the scientific community is still struggling to understand its origin. We therefore prefer to keep it.

Line 33 requires a reference.

The citation has now been added. See lines 37 in the MS with tracked changes

Line 48, I am not sure the waves were turning over because of counter measures alone. Susceptible population may have reduced as well, or may be local weather patterns etc

I think including statements of reduced susceptible population and local weather patterns is entering into speculative realm. We do agree with the reviewer that that the government counter measures may not have been enough on their own. We have altered the statement to include read...." These measures, and probably because of other reasons progressively reduced infections...." See lines 51 in the MS with tracked changes

The discussion has some of the results restated e.g line 295. This should be avoided for brevity.

We agree with the review. We have re-written the sentence to read... The B.1.549 lineage, mostly associated with Kenyan sequences and likely emerged from local transmission events was the second most prevalent lineage by wave two. See lines 299 in the MS with tracked changes

Line 328, Delta variant is B.1.617.2 not B.1.167.2

We thank the reviewer for catching this error on lineage naming. The B.1.167.2 has now been changed to B.1.617.2. See lines 332 in the MS with tracked changes

Reviewer #4 (Remarks to the Author):

Comments ignored